

# Low Water Stage Marks on Hunger Stones: Verification  for the
# Elbe River in 1616-2015
Libor Elleder [1], Ladislav Kašpárek [2], Jolana Šírová [3] and Tomáš Kabelka.[4]
[1] Applied Hydrological Research Department, Czech Hydrometeorological Institute, Prague, Czech Republic.
[2] T. G. Masaryk Water Research Institute, p. r. i., Department of Hydrology, Prague, Czech Republic.
[3] Hydrological Database and Water Balance, Czech Hydrometeorological Institute, Prague, Czech Republic.
[4] Prague Regional Office, Department of Hydrology, Czech Hydrometeorological Institute, Prague, Czech
Republic.
*Correspondence to*: Libor Elleder (libor.elleder@chmi.cz)
**Abstract**
The paper deals with the issue of documenting hydrological drought with the help of drought marks
(DMs) which have been preserved on dozens of hunger stones in the river channel of the Elbe in
Bohemia and Saxony. So far, the hunger stones have been regarded rather as an illustration of dry
seasons. Our aim was, among other issues, to draw attention to the much greater documentary value of
hunger stones and individual dry year marks inscribed on them. Therefore, we wanted to verify their
reliability and better understand the motivation of their authors. For this purpose, we used the current
extreme drought period of 2014-2019 which allowed detailed documentation of hunger stone in Děčín
with marks from 1536 to 2003. Thanks to the helpful position of the object near the water gauge, we
could compare the measured mark heights with the corresponding water levels. Simultaneously, we
have scanned the object into 3D format so that it is possible to perform a detailed inspection of all
marks, even those that were overlooked during field survey. A review of scientific and technical
literature from the 19[th] century showed that marks of low water levels on stones and rock outcrops
were to some extent interconnected with other important points. They were linked to zero points of
water gauges, initially set up for navigation purposes, and also to flood marks. A particular situation in
Děčín is therefore a unique example of epigraphic indication of low and high water levels in the
enclosing profile of the upper part of the Elbe river basin. To verify the marks of low water levels we
used the then current scientific studies which in the past brought the identification of dry periods.
However, we also used the oldest series of daily water levels measured in Magdeburg, Dresden, and
Prague, available by 1851, i.e. by the beginning of measurements in Děčín. These series had to be
reconstructed or digitized from the CHMI archive sources. Since 1851 we have been able to accurately
identify the heights and sometimes even the specific days when the minima were marked.
After thorough examination of field and newly measured data, as well as data obtained from review of
older literature presenting the first surveys of marks on hunger stones already in 1842, older marks of
low water levels can be considered mostly as a reliable indication of annual water level minima. The
aim of the mark creators was not to make the commemorative inscription on drought, but to register
the exact position of the water mark of the annual minimum. The deviations of most of the marks from
the water gauge records did not exceed 4 cm, in worse cases 8 cm and only exceptionally the disparity
was greater.
From the material obtained so far, the overall slight downward trend of minima since the end of the
18[th] century is noticeable. The view on minima of the 17[th] and 16[th] century is based on only a few data
and it is difficult to generalize so far. However, the minima obtained are comparable to or lower than
the data from the critical dry periods of 1842, and 1858 to 1874. Our verification and certain
rehabilitation of low water level marks should be an incentive to process all available epigraphic





documents of this kind in the near future in closer cooperation with colleagues from Saxony. The
potential of these objects offers a deeper knowledge of periods of hydrological drought and possibly
morphological changes in the Elbe riverbed.
**1.  Introduction**

In recent years, the phenomenon of drought has become the most prominent manifestation of climate
change in Central Europe. However, its objective evaluation and the evaluation of its extremity is
often a problem. The reason consists in difficult to grasp the phenomenon of drought or varying
impacts of it, respectively. Drought alongside the floods, though, rank among the most commonly
evaluated hydrological extremes. While the flood is caused by an unexpected and short-term excess of
water that causes damage, hydrological drought follows long-term deepening of water scarcity.
Our contribution is focused on hydrological drought, more precisely on minima of water stage of
surface water streams. The low water level and flow rate after long periods of deficit precipitation
represent particularly valuable information about the basin runoff. Therefore, they also report on the
base-flow, the groundwater accumulation, long-term depletion and hydrological drought propagation
(van Loon, 2015). The minimum water level or flow is, to a large extent, summary information on the
status of a given river basin.
Like floods, hydrological drought is difficult to study without examination of historical events.
However, what options do we have regarding low water levels? The available hydrological series
usually cover not more than 150 years. The longest hydrological series of measurements in Cairo 622-
1933, representing 1311 years of Nile observation (Shanin, 1985), was used to assess drought and its
interrelations with phenomena such as El Nino. In Europe, the longest series comprising
measurements of water levels in Magdeburg started in 1726 (see the following text), and the
measurements in Paris that started in 1731 (Delametherie, 1800). However, it is not possible to
conceal another complication, namely the later beginning of systematic hydrometric measurements
which are mostly available only since the end of the 19[th] century. This makes it difficult to estimate
flow rates somewhere. Therefore, stable profiles where we can assume the validity of the rating curve
as far back as possible are very valuable. Systematic series of water stages are therefore testimony on
runoff fluctuations but partly also on changes in the stream cross-section both natural and
anthropogenic.
Studies that focus on the identification of past dry periods and possibly on the wider context within
NAO, ENSO oscillations (e.g. Mikšovský et al., 2019) are mostly based on an analysis of precipitation
deficit or indicators that include temperature and hence loss by evaporation. They are necessarily
based on previous reconstructions of temperatures and precipitation based on an analysis of
documentary sources. However, if we want to describe how the rainfall deficits and other weather
influences were reflected in the runoff from the surveyed river basin, we have the options so far rather
limited.
Based on the available series of daily flow rates in Děčín (1851-2015), Brazdil et al. (2015) referred to
a period of low flows between 1858 and 1875. With the help of deficit volume analysis with fixed
annual ($Q_{95}$) and variable monthly threshold ($Q_{95m}$), they pointed out to drought corresponding to the
1904, 1911 or 1947 dry periods. The authors elaborated in detail selected dry years 1808, 1809, 1811,
1826, 1834, 1842, 1863, 1868, 1904, 1911, 1921, 1934, 1947, 1953, 1959 and 2003, i.e. 8 cases in
each century representing a total of 16 cases selected on the basis of the lowest Z-index and SPI1
values out of 10 homogenized precipitation series (Brázdil et al., 2012). Evaluation of particular years
includes meteorological and synoptic conditions, drought impacts, monthly values of air temperature,
precipitation, SPI1, SPEI1 and Z-index. Concerning the identification of the hydrological drought in
the 1860s and 1870s, a similar result was reached by Elleder et al. (2019) when analysing the
catastrophically dry year 1874 by analysing the newly reconstructed series of water levels in Prague
96  (1825-1890).





But what are credible documents on low water levels and a possibility of obtaining objective information on runoff before 1851, 1825 or even before 1726? Based on reconstructed data on temperatures and precipitation between 1766 and 2015, Hanel et al. (2018) indicated extreme deficits in precipitation, runoff and in water content of the soil surface layer. With regard to the affected areas, they identified droughts in 1858-1859, 1921-1922 and 1953-54 as extreme.

However, there is no doubt, similar to flood analysis, that verifying the model results according to the actual water level and flow rate considerably increases their credibility. We have a relatively large range of paleostage indicators to describe the maximum water levels during a flood. These are various types of shallow-water sediments, dendrochronological symptoms such as damage to trees, cave sediments, etc. (Benito et al, 2006, 2015). However, similar methods for estimating low water levels and flow rates are difficult to conceive. Therefore, only low water level indicators available through documentary sources remain (see Brázdil et al, 2018 for documentary data and the study of past drought, especially for epigraphic documentation). During the drought, attention was paid to objects normally hidden below the water level. Most often these were large boulders, protruding rocks, sometimes even point bars or slip-off slope sandy deposits with specific local names. In many cases these were also artificial objects, protruding foundations of old bridges and building elements; around the Rhine these were the remains of Roman buildings or old bridges, etc. (Wittman, 1859). Sometimes there was an interesting local tradition, in the sandstone area on the Czech/Saxon border it was the making of commemorative inscriptions, particularly inscribing the current year with low water level. Today, these objects are mostly called the hunger stones.

This article focuses on them wishing to clarify their purpose, origin and meaning. Traditionally, water management experts and historians and perhaps ethnographers in Bohemia considered inscriptions and year indication on hunger stones to be an interesting phenomenon symbolizing drought. At the same time, however, the understanding prevailed that the marks of "dry years" were merely commemorative records with no deeper meaning and that they were more or less randomly positioned. We believe that it is in this area that we have taken a substantial step forward in the explanation and possible use of these records.

We have therefore focused on the Děčín city located in the lower section of the Czech part of the Elbe river basin. The most well-known hunger stone is located here and all important height surveying of all the signs were carried out in the summer of 2015. In 2018 the whole stone was scanned. This article discusses to what extent the inscription years have the character of a historical minimum water level.

Objectives

1. To document and explain in more detail the phenomenon of hunger stones.
2. When are the year-marks only commemorative for that dry year and when do they represent the exact records of the annual minimum water levels?
3. Are there apparent relations in the heights of minima on different stones?
4. What is the relation to the systematic series of measurements?
5. Do the elevations suggest any trend in water levels?

## 2. Described region Czech-Saxon Switzerland and Děčín town

The Elbe river valley between Litoměřice and Pirna was made famous by a number of prints and paintings by 19[th] century romantic painters such as Adrian Zinggs (1734 – 1816) and Caspar David Friedrich (1774 – 1840). A. Zinngs born as a Swiss, who lived in Dresden, probably coined the name of the region Saxon Switzerland, and later extended to the Czech — Saxon Switzerland (Frölich – Schauseil, A., 2018). The Elbe, which leaves the territory of the Czech Republic in the deep rocky canyon and ends here its upper stretch, flows between Lovosice and Děčín through the Krušné hory mountain system. Along its path it first intersects the volcanic zone of the České středohoří area. Below Děčín, it then flows through the landscape of sandstone rock formations. The Elbe riverbed is situated at an altitude of about 120 m a. s. l. in a deep sandstone valley 200-300 m below the level of




the sandstone plateau (350-450 m a. s. l.). Protruding volcanic formations reach a height of 500-800 m
a. s. l. The Děčín and Hřensko cross-sections represent the closing profiles of the Czech part of the
Elbe. In addition to wood, the local sandstone was a traditional building and sculptural material here
and throughout the North Bohemian region. However, it was also used for rich epigraphic production
on the spot — on rocks and boulders (Jenč, P., Peša, V., Barus, M. 2008). It is quite logical that water
levels were recorded at river where possible, both minima and maxima.

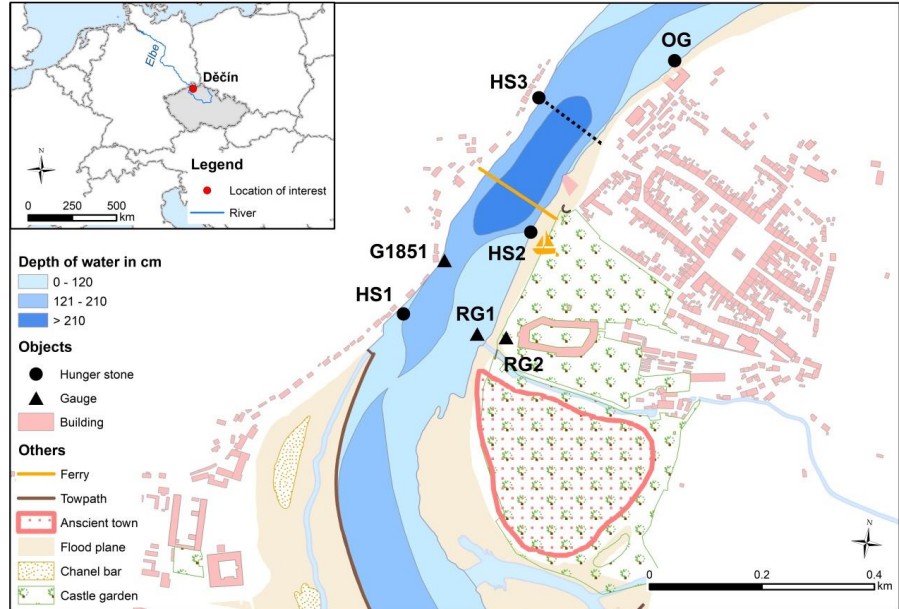


*Fig. 1 The Děčín city in 1842 with indication of the original extinct town (13th — 14th century), area of*
*shallows (the lightest blue), water gauges RG1, RG2, G1851 and OG and three hunger stones (HS1,*
*HS2, HS3)*

At the centre of our study is the Děčín city (Fig. 1) known among other things for its unique series of
flood marks (Brázdil et al., 2005, Elleder, 2016a) and by just explored hunger stone. The earlier
documentation (see the following text) which comes from commission inspections of the Elbe
riverbed revealed previously unknown facts. In 1842, there were still in total three hunger stones in the
Děčín city with engraved years, two on the left [HS1, HS3] and one on the right bank upstream the
ferry [HS2] (Protokoll, 1842). The preserved stone [HS3] which is located in the lower part of the
deeper riverbed is in the centre of our attention.
There were at least two places in Děčín that were problematic from the navigation point of view. The
first hunger stone [HS1] was located near the first water shallows area. It is related to the confluence
of the Elbe River with the Ploučnice River from the right, the Jílovský stream from the left and
sediment deposits. This place with a ford at the confluence and below the protruding sandstone ridge
was probably advantageous long ago as a settlement. At the end of the 13th century a royal town was
founded here, Fig.1, (Velimský, 1991). Possibly in connection with the period of a significant
occurrence of floods between 1342 and 1374 (Elleder, 2015) it was abandoned and transferred as a
serf city to the other side of the rock ridge where a castle stood and nowadays the manor house is
situated. On the rock under the castle there are flood marks from 1432 carved into the rock block.
Alongside, a water gauge is located with indication of the Prague ell units of length (59 cm) [RG2].





This gauge starts at 9 ells above the water level for full navigability (Bohemia daily, 1845). This depth
was traditionally referred to as the "Fünfspanner", i.e. "five-span", a sufficient navigational depth of 5
spans or 50 inches, or 125-130 cm for the full loading of the Elbe ships (Bohemia daily, n. 45, from
April 4th, 1845). There was a rock block near the shore with a water gauge for low water levels in feet
[RG1] (1 to 5 feet), probably related to safe passage. In 1851, water levels in Děčín began to be
systematically monitored, initially at the old water gauge [OG] at the site of the navigation directorate.
Apparently, the water gauge served the navigation to efficient ship loading for the place of the second
water shallows area. It still bears the original German, now popular, name "Heger", or supervision.
Later, the observation was transferred to a new water gauge [G1851] (see chapters on methodology,
documentary sources).
**3.   Methodology**

**3.1. Data and documentary sources**
The first partial goal was to prove that the water level marks on the hunger stone in Děčín and other
stones were meant by their creators as signs of annual minima in the years attached to the line. The
simplest means is a comparison with concurrent water level measurements on a near water gauge
(accurate identification) and also use of other available measurements (approximate confirmation of
significant water level decline). We mainly used four series stored in CHMI (Czech
Hydrometeorological Institute). These are the systematic series at sites of Magdeburg (1726-1880),
Dresden (1801-1829), Prague (1825-1890) and Děčín (1851-2019).
**3.2.  A series of daily water levels in Magdeburg 1726-1880**
Around 1880, this series was acquired by Prof. Harlacher (Elleder, 2012) from the Water Management
Directorate in Magdeburg. It was found 110 years later in the unclassified funds of the Hydrological
Service in the 1990s. A copy was sent to the IKSE Magdeburg headquarters. Its digitization was
carried out in 2005-2007 in cooperation with CHMI and T. G. M. WRI. The value of these
measurements is considerable as the series covers continuously the whole period of 64 years of the
18th century having no other alternative for Central Europe. Its disadvantage is the downward trend in
annual minima which can be explained largely by shortening, deepening and changing the profile of
the Elbe river around 1816 (Simon et al., 2005). However, in our case we can identify very well
particular annual water level minima and their association with the years on hunger stones between
1746 and 1800 (hereinafter DM for the minimum water level signs). By identifying the annual
minimum water level in Magdeburg, we could estimate the likely date of making DM in Děčín
considering Děčín-Magdeburg water transit time (6 days).

**3.3. A series of daily water levels in Dresden 1801-1829**
A copy of this series made probably by an official of the Prague City Hall in 1829 offers an evidence
that the systematic series does not begin in 1806 (Fügner, D., Schirpke, H., 1984, Fügner 1990) but at
least in 1801. The series was found in the 1990s by a private researcher J. Svoboda in the Archive of
the Prague capital, and he left it to CHMI. Dresden has a clear advantage over Magdeburg in its
geographical proximity to Děčín, so we preferred it for the period 1801-1829.

**3.4. A series of daily water levels in Prague 1825-1890**
In Prague, an occasional water gauge (possibly flood gauge) was probably established by the director
of the Klementinum observatory A. Strnad in the profile at the Monastery of the Knights of the Cross
in 1782 (Brázdil et al., 2005, Elleder, 2016a). Later (in about 1821) it was transferred to the profile of
the Old Town mills. Systematic observation of the water gauge started in 1825 (in more detail,
Elleder, 2016). The profile of the Old Town mills was related to the weir normal (i.e. to the weir crest)





so it was a profile that did not change. According to Novotný (1963), the original observation diaries
and perhaps even annual reports of measurements were lost. Only the published values of monthly
minima, maxima and averages in the yearbooks of the Klementinum observatory remained. In
similarity with other observations (e.g. in Magdeburg, or in Vienna), Prague observations were
published weekly and later daily in daily newspapers. Therefore, we decided to regain daily
measurements of water levels published in the daily Prager Zeitung starting with January 1825. The
data were collected for three years by an external CHMI associate Mr. Zvonimír Dragoun in the
archive of journals and newspapers of the National Museum in Prague. The measurements were used
similarly to the previous series particularly for the period 1825-1850. A special publication will be
devoted to the complete time series.
**3.5. A series of daily water levels in Děčín 1851-2019**
In similarity with other profiles along the Czech section of the Elbe River, a systematic observation of
water levels was introduced in Děčín. At first there was an old water gauge [OG] (Fig. 1) which was
located in the profile at the site of the steamship navigation directorate probably before 1842. Later,
but probably not earlier than from 1858, the new water gauge [G1851] started to be used on the pillar
of the Empress Elizabeth Bridge (built in 1851). The problem is a newly found uncertainty in the
change of the zero point of the water gauge (Protokoll, 1858) whose height might have been elevated
in 1858 by 16″ (i.e. by about 42 cm). It is not entirely clear from when exactly the data from the old
annual reports of measurements (monthly reports are available only after 1875) of the Děčín series are
related to the new zero height. Minima of water levels on hunger stones [DM] are therefore partly a
possible verification of early measurements in Děčín. Even later, around 1877, the water gauge was
transferred to the waterfront (Harlacher, 1883). At that time from November 1876 to March 1881,
prof. A. R. Harlacher was performing hydrometric measurements with his colleague J. Richter and
associates (Harlacher, 1883). From this time, we have measurements up to 169 cm of water level at a
measured flow rate of 90 $m^3 \cdot s^{-1}$ (Tab. 1). For interpolation and extrapolation of the curve, the formula
$Q = 78.09 (H_0 + 1.45)^{1.953}$ was applied. According to this formula, the water level at 140 cm ($H_0 = -60$
cm) would correspond to a flow rate of 57 $m^3 \cdot s^{-1}$ ($H_0$ is the water level corresponding to the height of
the water before the shift of zero of the water gauge by -200 cm made on 1 October 1939). Novotný
(1963) reports the successive shift of the rating curve and presents the evaluation of historical flow
minima. Of these, for the water level of 133 cm (on 23 August 1868) he reports the flow rate of 50
$m^3 \cdot s^{-1}$ and for the stage of 137 cm (on 6 September 1874) the value of 54 $m^3 \cdot s^{-1}$. After the river bed
modification around 1891, the curve changed substantially in the section of low flow rates (Tab. 1).
He evaluated the significantly lower flow rate for the water stage at 113 cm only later on 19 August
1904, at 39 $m^3 \cdot s^{-1}$. This is a significant difference that would affect the flow rates at the extreme
minima of 1868 and 1904, and the question is whether to trust the 1876-1881 curve when it was not
possible to evaluate the lowest water levels because they did not occur. Hydrometry of small flow
rates on the Saxon side has been available since 1886, but extremes only since 1893. Therefore, in the
results, the flow rates at individual minima are accepted so far with caution.

Tab. 1 The oldest measurements of very low flow rates in Děčín and on the Saxon side

|  | Date | H [cm] | Q [$m^3.s^{-1}$] / location of the flow rate measurement |
|---|---|---|---|
| **Děčín** | 28. 7. 1876 | 163 | 90 / Děčín * |
|  | 13. 10. 1877 | 167 | 96 / Děčín* |
|  | 29. 8. 1893 | 144 | 63 / Děčín* |
|  | 13. 8. 1904 | 119 | 46 / Děčín* |
|  | 29. 8. 1911 | 118 | 56 / Děčín* |
| **Dresden** | 17. 7. 1893 | -179 | 56 / Großschepa ** |
|  | 14. 7. 1893 | -172 | 63/ Kötschenbroda ** |

*Old hydrometry, 1877-1940, **( Elbeströmbauvervaltung, 1897)



### 3.6. Preliminary verification of the heights of assumed water level minima using regional press and simultaneous measurements on the existing water gauge in Děčín

This study was preceded by about 10 years waiting (since 2005) for a suitable opportunity to undertake a field survey of hunger stones that are totally or partially below the surface at normal summer flow rates. There was no other possibility than to try to find an alternative solution. In 2009, as part of a preliminary study, we tried to use rich iconographic material from the period 1894 to 1994 and reports of hunger stone in Děčín in contemporary newspapers. In the older press materials, reports were looked up when the hunger stone was visible and an indication was given as well which year-marks were above the relevant water level. Then it was easy to classify the marks into height groups with a water level higher than that of the day reported. Further specification of heights was possible only on the basis of photographs by comparing which mark is higher or lower in the given group. The marks were connected by contour lines indicating the resulting bands. Estimated water levels were then compared with annual minimum values. The result pointed to the expected possible concordance with annual water level minima. We have followed a partially similar approach with the hunger stone in Pirna.

### 3.7. Field measurements

In 2011, it was possible to carry out a field verification of the estimated heights of the marks that were located in the highest part of the stone. In 2014, the opportunity was not used as we believed that the dry season will have a longer-term character which was confirmed in 2015 and 2018. In 2015, the hunger stone in Děčín [HS3] and the stone in Těchlovice were surveyed. During the surveying of the stone in Těchlovice located on the slip-off slope gravel deposits, it was not necessary to make any ground adjustments. However, only relative heights recalculated to the minimum height of 1842 were measured.

The surveying of the Děčín stone in 2015 required preparation representing sediment removal and stone cleaning (manual work of 2 to 3 people for 3 hours or more). In 2015, the sediment layer reached the sign of 1616, i.e. around 70 cm of height. In addition, it was necessary to make a pit around the stone's very low marks. The use of a pump with a syringe to wash away sediment, blasting stone and pumping water from the sump significantly accelerated the work.

The measured mark heights were linked to the nearby fixed geodetic point. All surveyed geodetic levelling points were photographed. The measurement took place on 14th August when water levels dropped the lowest just before the expected rainfall episode which increased the Elbe water level significantly. Participants of the measurements were: Ladislav Kašpárek and Jan Kašpárek from T. G. M. WRI, Libor Elleder from CHMI and a land surveyor, Mr. Zvonimír Dragoun (presented on EGU 2016, Elleder, 2016b).

We basically did the same when scanning and creating a 3D model in 2018. The stone was prepared by colleagues from CHMI: Martin Groušl, František Pěkný, Martin Hubený in advance on 27th July. The final adjustment was made on the day of measurement and was assisted by Daniel Kurka, Libor Elleder and Martin Hubený. Martin Hubený also performed a hydrometric measurement in the hunger stone profile (Fig. 1 [HS3]) including the cross-section measurement using the ADCP (acoustic Doppler current profiler). 3D scanning was performed by Libor Tělupil from the VR3D Company (http://vr3d.cz) on 30th July, lasting for about 3 to 4 hours. Similarly, the marks on the rock in the [RG2] profile were scanned. Because scanning requires soft, shadow-free lighting, a temporary stand was placed over the stone. The whole event was documented by the local press (https://www.idnes.cz/usti/zpravy/decin-vodomer-hladovy-kamen-skenovani-3d-model.A180730_113803_usti-zpravy_mi) and the result is partially accessible on the CHMI website (http://portal.chmi.cz/historicka-data/hydrologie/zaznamy-z-minulosti/hladovy-kamen). Both measurements in 2015 and 2018 were performed during hot summer days with temperatures in the first case 38 °C, in the second case around 30 °C. An independent surveying campaign was carried out in 2015 by the Elbe River Administration, state enterprise (Randák et al. 2015, 2018a, b) and in 2018 also by hydrologists and archaeologists from Saxony (Walther et al., 2018).



### 3.8. Measurement processing

In 2015, 33 points were surveyed, mostly engraved lines with attached year indications. For obvious reasons, making a DM mark is much more difficult than making a flood mark. It is difficult to estimate when the water level starts to rise (see discussion). Therefore, it was not always certain whether the sign would represent an indication of the immediate low water stage (LL), the local minimum (LM) or the annual minimum (AM). For verification and approximate determination of the minima marked on hunger stones (DM) until 1726, there are only available documentary sources, i.e. reports on weather and impacts of hydrological drought, such as drying of smaller streams, wells, shutdowns of small and medium mills, or a necessity to travel to a grain mill tens of kilometres. We reproduce this information primarily from Brázdil et al. (2015). The decade frequencies of drought occurrence since 1500 (Brázdil et al., 2013) were a valuable basis for verifying the position of marks, especially in the 16th and 17th century.

For the evaluation of the DM marks made after 1727 we used the above-mentioned series of measurements in using the Magdeburg series rather for dating verification and the Prague and Dresden series for assuming a very approximate estimate of the significance of the minimum. Concerning newer cases after 1851, it is possible to confirm the correct or incorrect position of the mark (DM). Regarding deviations from the measured water level for that day, we consider the precisely marked height (HP) at a deviation of 0-4cm and approximately marked height (CP) at a deviation of 4-8 cm. We consider larger deviations as a possible mistake when placing the measuring rod or a poor understanding of the difficult-to-read position of the mark or line. If the DM mark does not have an accurate dating, we can assume dating according to the minimum water level when there is the exact (HP) identification with the minimum water level.

A very important product is the digital model of a hunger stone, which can be viewed, edited in contrasts by selecting the option *"shaders"* using the Meshlab processing system (http://www.meshlab.net/), and thus clarify the unclear situation and illegible marks. Because at the time of measurement we had not always understood the situation in situ, it was possible to derive the missing height from the digital model by reading the position (x, y, z). Thus, the second mark was found on DM1616, DM1536, etc. In the survey diary, the actual measurement is clearly arranged, documented by photographing of the position of the measuring rod and by the highlighted view of the described part of the stone. The measured heights of all marks and the position are presented on the stone which is divided into 4 height zones and the embankment side [ES], left side [LS], right side [RS], front platform [P] and the highest parts of the ridge [R]. The presentation of the marks is chronological, so that the information is combined into a logical complex.

### 3.9. Complementing measurements according to other objects

Some marks (DM) are missing on the Děčín stone, but we find them elsewhere. If their heights were measured during commission inspections of the Elbe River in 1842 (Protokoll, 1842) and 1850 (Protokoll, 1850), relative to the level of 1842, these differences can be utilized. Thus, some heights of extinct stone [HS1] were added in Děčín (1766, 1782), Dolní Žleb (1516, 1615, 1636, 1706, 1834 and 1835) and Pirna (1706, 1834 and 1835). For other hunger stones, we can only take into account the position of the marks, reviewing whether it is in accordance with or contrary to the facts found.

### 4. Results

### 4.1. Brief history of low water stage records in context

4.1.1. The oldest documented field surveys of the Czech rivers 1640-1726 and trends in water levels

It is very likely that the most objective records of hydrological drought or more specifically records of low water levels are related to navigation in Central Europe (Brázdil et al. 2019b mentioned limiting of water transport in 1686 and 1746 years). It cannot be ruled out, for example, that mapping of the Vltava River (by David Altmann of Eidenburg) and the river regulation by Kryšpín Fuk (1640-1643, abbot of the Premonstratensian monastery in Strahov), (Wiesenfeld, 1844) were made possible just by



a drier period, probably culminating in 1642 (documented by Pekař, 1998). Also, surveys of the upper
Vltava river reaches carried out by Lothar Vogelmonte for the intended canal between the Danube and
the Vltava rivers in the years 1700-1715 show a possible time relationship (Wiesenfeld, 1844). The
dry years 1705, 1706 and 1707 (marked on hunger stones) could be an opportunity to explore the
streams in times of low water level. Drought in 1726-1728 clearly affected the beginning of water
level measurement in Magdeburg (Hofmann, 1850). It was probably connected with the frequently
quoted commission of Jan Ferdinand Schor that carried out a survey of the Vltava River with regard to
navigation and the construction of the first lock chambers (Wiesenfeld, 1844). The agreement on duty-
free navigation on the Elbe (see Faulhaber, 2000, 2013) from 1821 (the year was also marked on the
stone in Děčín [HS3]) along the Elbe river up to Hamburg led to increased interest in monitoring water
levels for individual participating states including the Austrian Empire, Saxony up to Denmark.

The catastrophic dry period of 1834 to 1836 affecting both the Elbe and the Rhine basins raised the
issue of a general downward trend in water levels, especially in the Elbe basin. H. Berghaus pointed
out this trend and the poor prospects of the Elbe navigation (Berghaus 1836, 1854). A forestry expert,
Prof. Reuter of Aschaffenburg (Reuter, 1840), pointed out the possibility of this trend being linked to
deforestation of the Central European landscape.

4.1.2. The Elbe Commission in 1842 and surveying of hunger stones

In this context, there is a link with the disastrous dry year of 1842 (Brázdil et al, 2019a indicated that
in 1842 summer precipitation was significantly reduced from western to eastern central Europe) and
the Commission of the Elbe states (Austria, Saxony, Prussia, Anhalt, Hamburg and Denmark)
organized to improve navigation conditions. The aim was a thorough description of all fixed points
(water stage gauges, flood marks and also marks on hunger stones), navigation conditions and
minimum navigation depths along the navigable section of the Elbe from Mělník town to Cuxhaven.
Stones and rocks in the river were of double importance for navigation. They were a dangerous
element, but at the same time they served the orientation for navigation. The commissioners travelled
by boat and the section Mělnik-Meissen was surveyed from 5th to 11th September 1842, 14 days after
reaching an absolute minimum water level. The water levels of the Vltava and Elbe were still very
low, but they were already 9 to 20 cm higher than the minimum in the previous August. In Děčín
town, measurements were made from 7th to 8th September (Protokoll, 1842) at a water level at about
3.5″ (9 cm) above the 1842 minimum. Three hunger stones in Děčín (Fig. 1) and one in Dolní Žleb
were identified and surveyed. On the Czech side, a water gauge in Litoměřice and a water gauge for
navigation purposes in Děčín were noted in the section between Mělník and the state border (in both
cases there were no regular records available). On the Saxon side, water gauges in Bad Schandau,
Pirna, Dresden, Meissen and Riese were identified, managed by the Royal Navigation Directorate
(Königl. Wassebaudirection Dresden). The hunger stones were detected and partially surveyed in the
following locations: Schmilka and Pirna (see the text below), (Protokoll, 1842).

4.1.3. The Elbe Commission in 1850 and making a link of the water level minima to the flood marks

The Commission compared the situation with the last commission survey in 1842 and registered the
removal of some barriers to navigation. Gauging some low water levels through their relation to fixed
points is of utmost importance to the subject of this study. These fixed points were only flood marks
(Roudnice, Ústí nad Labem, Děčín), and alternatively the current water level in 1850, or zero point of
a water gauge were used (old water gauge in Litoměřice, Ústí nad Labem, railway water gauge in
Dolní Žleb, water gauge during the surveillance in Pirna). Until now, only two of the original three
hunger stones remained in Děčín. The Austrian Commissioner carried out a precise survey of all the
flood marks on the castle rock in Děčín (Krolmus, 1845, Brázdil et al. 2005) and related their heights
to the minimum of 1842. The Commission was active in September when there was a significantly
higher water level than in 1842. Therefore, the marks on hunger stones were underwater and so
difficult to recognize. For the present stone [HS3], its top at 14½″ (37.7 cm) was below the then
current water level. Since, according to our measurement, the top is at the water level H = 176 cm, the
then current water level was about 214 cm and the flow rate was about 190 m³·s⁻¹ (according to
Harlacher's rating curve, 1883). The Commission had a new map of the Vltava River and the Czech
Elbe river which was created between 1843-1848 (Elbekarte, 1848) with already marked depths in





cross sections. In the following year on 1ˢᵗ January 1851, the daily observation of water gauges on the
Czech Elbe River in the Mělník, Roudnice, Litoměřice, Ústí nad Labem, Děčín, and probably Dolní
Žleb cities begins. Zero points of the new gauges were established 6 inches above the minima in 1842
(Protokoll, 1858). At this stage, half-cargo navigation was possible (Wex, 1873).
4.1.4. The Commission and the catastrophic drought of 1858
The year 1857 was very dry, just like 1858. The commission was in Děčín on 20ᵗʰ May 1858. The
water level was in the range of -0.75 to -2.5″ (about -2 to -7 cm) according to of the new water gauge.
Just before that, according to the Protokoll (1858), the height of the zero point of the water gauge in
Děčín and Dolní Žleb was increased by 16″ (42 cm). The Commission identified the 1857 minima as
generally the lowest in the period between 1842 and 1858.
Considering the record low water levels of the Rhine, Dr. Josef Wittmann, Director of the Society for
the Study of the History and Monuments of the Rhineland, published a comprehensive publication
(Wittmann, 1859) which is also an inventory of periods with low water levels of the Rhine from 70
AD to 1858 and an overview of prominent objects hidden under normal Rhine water stage under
water. According to his work, the level of the Rhine dropped record-deep in 1858, deeper than in
1788, 1813, 1818, 1822 and 1830, at least according to the water gauge in Cologne. It was this
alarming water level that was both the main motivation and the opportunity for his work. The year
1858 was recently indicated by Hanel et al. (2018) as one of the most extensive drought periods. The
years 1857 and 1858 in the Elbe basin are also at the beginning of two decades with the occurrence of
significant and catastrophic periods of low water levels. These are the years 1858, 1863, 1864, 1865,
1868, 1873 and 1874 (Elleder et al., 2019) most of which can be found on various hunger stones in the
Elbe. Professor Bruhus of Leipzig (Bruhus, 1865) was at that time studying hydrological drought in
Saxony. His work was a basis of a study by a Forest Counsellor von Berg (Berg von, 1867) which
presents again the same idea of the loss of water in all Central Europe and documents it with the help
of precipitation balance and minimum water levels not only the Elbe, Oder and Rhine, but also the
Elster and Mulda rivers. The author saw the cause again in the intensive use of the landscape,
especially in its deforestation. The prominent Austrian water manager G. von Wex (Wex, 1873)
applied the recorded minima of water levels from 1616-1842 when demonstrating a steady downward
trend in 1842-1873. He also recalled the earlier views of H. Berghaus or the Prussian Counsellor
Hagen. However, Hagen refuted the downward trend for example for the Rhine. A noted expert in
hydrometry, H. Grebenau, participating also in the famous international survey of the Rhine in 1867,
on the other hand, supported the idea of flow decline with his flow measurements.
This drought also had a specific impact in the most industrial part of the Austrian monarchy, Bohemia.
In 1869, another Elbe Navigation Commission (Wex, 1873) was held. In 1871, Professor of the Prague
Technical University, A. R. Harlacher, established a temporary station for hydrometric observations
and calculating the amount of runoff from the Czech Elbe (1871-1872), (Harlacher, 1871, 1872). The
year 1873 brought, according to Cvrk (1994), the intensification of river regulation of the lower Elbe
(mostly digging and removing boulders) and finally deepening of the riverbed by approx. 20-30 cm.
The catastrophic drought in 1874 led, after a broad discussion, to the establishment of the
Hydrographic Commission of the Kingdom of Bohemia based in Prague (Elleder et al., 2019). The
floods and the generally wetter period of 1880-1882 ended the long 1858-1878 period with the
occurrence of drought. Extensive hydrometric measurements including a detailed mapping of the
riverbed were made by Harlacher in Děčín between the old road bridge and the railway bridge in the
1880s (Harlacher, 1883). Harlacher was interested, as H. Berghaus earlier and G. von Wex at that
time, in the downward trend of the Elbe water levels. Therefore, he collected the above mentioned
series of measurements (Dresden series 1806-1872, not found in CHMI, and Magdeburg series 1727-
460  1880).

4.1.5. River regulation of the Elbe — earlier and so more often appearance of hunger stones
After the period from 1880 to 1891, the low water levels in 1892 and 1894 intensified the pressure to
regulate the Elbe. In 1896, a Canalization Commission was established for the regulation and
canalization of the Elbe between Mělník and Ústí nad Labem. The aim was to build a navigation link
up to Prague and ensure a navigation depth of 180 cm, an increase by 50 cm, in the period 1896-1938



(Cvrk, 1994). This is a very important fact for our work which resulted in a substantial shift of the
flow rating curve in the Děčín profile in the part of low flow rates by about 50 cm.
The next stage was putting into operation the Vltava cascade, the construction of the Slapy waterworks
in 1957 (https://www.kct-tabor.cz/gymta/VodniPrehrady/Slapy/index.htm) and the Orlík waterworks
in 1963 (https://www.kct-tabor.cz/gymta/VodniPrehrady/Orlik/index.htm). After this date, the minima
of flow rates are significantly higher than the previous ones (36 to 51 $m^3 \cdot s^{-1}$). In times of low water
levels, the flow rate is enhanced sometimes by even 20 to 30 $m^3 \cdot s^{-1}$. For this reason, the flow minima
are today around 65 to 75 $m^3 \cdot s^{-1}$. This means that we need to divide the tags on HS3 into at least three
basic groups: a) 1516-1896, b) 1897-1956 and c) 1957-2018.
**4.2. Hunger stones and other indicators of low water stages in the European context**
4.2.1 Hunger stones, antique monuments and other indications of low water stages in the Rhine basin
Witman's work suggests that the oldest designation dates back to 1305 in Olten on the Aare River and
in Strasbourg in the same year or in 1302 or 1303. The most notable example is the so-called
"*Laufenstein*" in Laufenberg at the confluence of the Aare and the Rhine used to be visible if the
Rhine flow decreased below 300 $m^3 \cdot s^{-1}$. Civil Engineer Heinrich Walter surveyed the marks on this
stone around 1890 (Walter, 1901). There were a total of 10 DM marks: 1541, 1692, 1750, 1764, 1797,
1823, 1848, 1858, 1891, and 1893 (Walter, 1901 presented altitudes of 1541, 1750, 1858, 1891, and
1893). Some marks were compared with the observed series and corrected by Pfister et al. (2006).
Near Unkel in the dry season of 1766, the dates of 1521, 1567 and 1639 were visible on the basalt rock
called *"Unkelstein"* (i.e. basalt in the Land of Rhineland-Palatinate in translation). However, the
situation in autumn 1766 was ¼ feet lower (Johannes Jansen notes, Weikinn, 2000). In the past, there
were several places in the Rhine basin known as *"Hungerstein"* or *"Hungerfelsen"*. One of the oldest
pictorial documents was published by Merian (Merian, 1645), perhaps according to the field sketches
of Prague graphic artist V. Hollar who after emigration cooperated with M. Merian. In the foreground
there is the so-called *"Ara Bakchi"*, *"Altarstein"* or *"Elfenstein"* (Fig. 2, Fig. 3)) which is just one of
the sites that used to be accessible only during the low water stages of the Rhine.

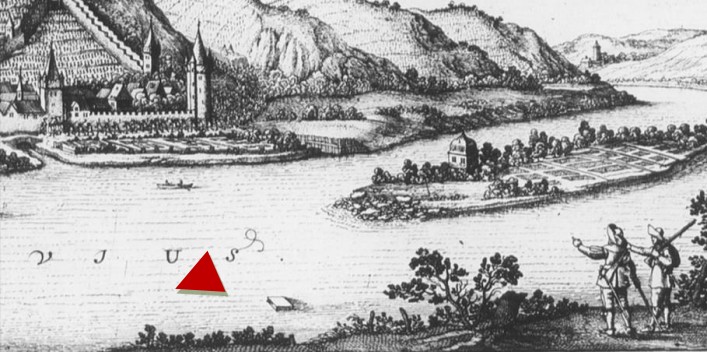

*Fig. 2 Ara Bakchi, Altarstein, Elfenstein near Bacharach, perhaps in the dry season of 1636, 1639 or 1642,*
*(Merian, 1645), the position of which is marked by a red triangle in a cut-out view of Bacharach.*

Among similar objects there is, for example, the rock in Olten in the Aare River. Around Bodensee,
such objects indicated low lake levels in Staad, Mammern and Konstanz. In 1750, the remains of the
assumed ancient buildings, the pillars of the bridge in Cologne and the aforementioned Altarstein were
visible during low water levels, and in 1746 the pillars of the old bridge in Mainz were visible (Fig. 3).
A tradition of storing 12 bottles of wine at a hunger stone on the bottom of the Moselle in Trarbach is
also remarkable.





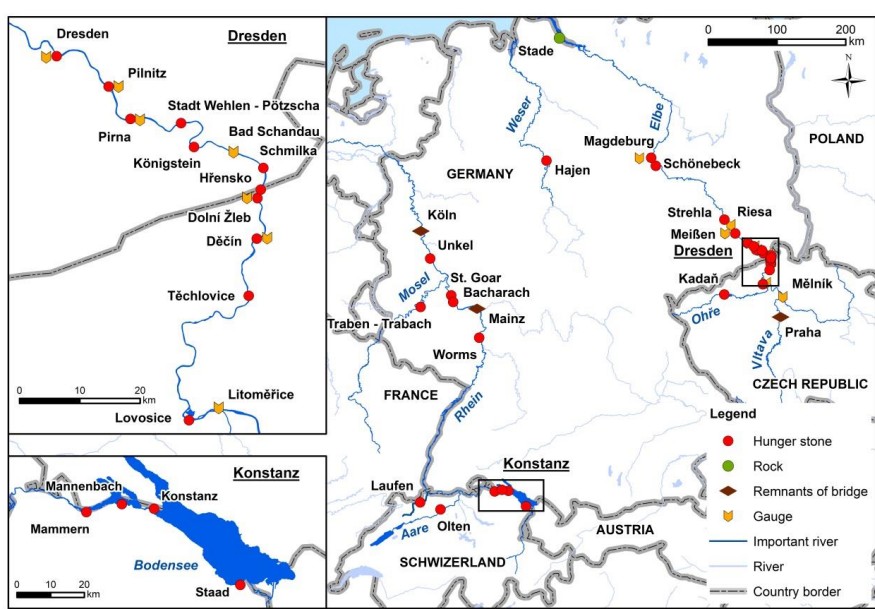


*Fig. 3 Central Europe and occurrence of objects similar to hunger stone in Děčín*

4.2.2. Hunger stones on the Elbe and their removal
The Elbe along its upper reaches is a much smaller river than the Rhine, for example, in the narrow
canyon area between Bingen and Koblenz (with average flow rate approx. 2000 $m^3 \cdot s^{-1}$, minimum
around 400 $m^3 \cdot s^{-1}$). The Elbe has an average flow rate of approx. 300 $m^3 \cdot s^{-1}$ between Děčín and Pirna
and without the enhancement by the Vltava cascade the minimum flow rate was dropping until 1957,
or 1963 respectively, as low as to approx. 35-41 $m^3 \cdot s^{-1}$ in the years 1904, 1911, 1921, 1934 and 1947
(Novotný, 1963). The lowest water levels were recorded on the Rhine in October or in winter. Low
water levels of the Elbe are the most common from June to September, but in 1874, for example,
lasted until December (Elleder et al., 2019). However, low levels are recorded in times of severe frost
even in winter. On the Czech side downstream, the first but rather modern stone was the object in
Lovosice (since 1904), then in Těchlovice (1 hunger stone, further HS object), Děčín (1-3 HS objects),
Dolní Žleb (11 HS objects), Hřensko (15 HS objects), Schmilka (1HS), Koningstein (2 HS objects),
Pirna (2 HS objects), Wehlen (1), Pillnitz (1 HS), Dresden (3 HS), Meissen (?) and Strehla (1 HS) (see
Fig. 3).
The term Hungerstein was not often used in the 19th century. In the scientific literature we find the
heights of the low water levels as *"Merkzeichen der Wasserstände"* (Neue Schriften, 1845), in the
news reports the term millstones as *"Malsteine"* appeared. The commissions' reports in the Protocol
(1842) and Protokoll (1850) mention the stones as *"Steine"*, and the remarkable ones as
*"Merkwürdige Steine"*. The Elbe in the sandstone canyon used to be rich in local names: *"Frog
Stones"* or *"Froschsteine"* (Dolní Žleb) (Protokoll, 1842, p. 44), as well as *"Monk's Stone"*
*(Mönchstein)* and *"Millstone" (Malstein)* that were removed already in 1858 (near to the customs
office in Dolní Žleb). Two hunger stones with dating (see the text below) opposite the church were
designed for blasting. In 1842, stones near Žertovice, and in 1850, on the Saxon side at the Ober
Vogelsang site (the so-called *"Hermsteins"*) were blasted away. The term Hungerstein appeared in a
newspaper article in 1842 (Pillnitz) in a newspaper in connection with HK in Meissen in 1865





(Rumburger Zeitung No. 47 of 11 October 1865), and in 1876 (Teplitzer Zeitung No. 98 of 30 August
1876). The Czech derived mutation " hladový kámen" ("hunger stone") was introduced by the regional
daily Jizeran (17 September 1892) during the drought in 1892.
4.2.3. Hunger stone in Těchlovice
The site is located above the sandstone canyon and the valley is formed by rocks of volcanic origin.
On the left bank of the Elbe River, approximately in the river km 85 (below Mělník), the Elbekarte
map (1848) shows the so-called *"Mändelstein"* in the riverbed, but actual stone is on a gravel bench
and the affinity of the objects is unlikely. The Protokoll (1842) mentions a strong current and a place
with a depth of 1′ 8″ that is only about 50 cm. The Protokoll (1850) only reports on depths around 160
cm, the Protokoll (1858) does not mention depths nor stones in Těchlovice at all. Estuaries of two
streams create flow sediment cones, during low water levels the stone is separated from the water and
lies on a wide gravel bench. For technical and time reasons, only relative geodetic link and height
measurements were made in 2015. There are 7 marks on a flat boulder of volcanic origin (1868, 1874,
1892, 1904, 1928, 1980, 2015), (Tab. 2). The mark of 2015 was prematurely made by an unknown
person and does not correspond accurately to the minimum water stage that occurred later.
Tab. 2 Survey of DM heights in Těchlovice

|      | $H_R$ heights | $H_{1842}$ | $H_{DE}$ |
|------|---------------|------------|----------|
| 1842 | -104          | 0          | *132*    |
| 1874 | -108          | -4         | *128*    |
| 1892 | -109          | -5         | *127*    |
| 1904 | -133          | -29        | *104*    |
| 1928 | -114          | -10        | *122*    |
| 1980 | -102          | 2          | *134*    |
| 2015 | -150          | -46        | *86*     |

***HR*** *water level of DM, levelling in 2015 linked to auxiliary point,* ***H₁₈₄₂*** *DM water level relative to the level of*
*DM1842,* ***H_DE*** *water level accommodation to present Děčín gauge, H_DE, approximate conversion to the water*
*level in Děčín according to water stage in 1842 (132 cm)*
4.2.4. Hunger stones in Děčín
In 1842, three hunger stones were examined within activities of the Elbe River Commission
(Protokoll, 1842), (Fig. 1).
According to the report, the first hunger stone [HS1] was located near the left bank of the Elbe
opposite to the castle rock, i.e. also opposite to the well-known flood marks of 1432-2013 and the
historical rock water gauge [RG2] on the right bank (Brázdil et al., 2005), (Fig.1). On the stone [HS1],
the approximate depths of [DM] minima in 1719 and 1766 were measured in September 1842. The
1782, 1790, 1835 and 1842 marks were surveyed precisely (Tab. 1). Elevation ratios were expressed
as heights above the previous August minimum of 1842. In 1850, the depth of the 1782 mark [HS1]
was determined as 7.5″ (19.5 cm) below the water level, the 1842 mark was not visible (it follows
from Table 1 that it was 41.5 cm below the water level). Protokoll (1850) implies a link of this mark
with a water gauge for low water levels [RG1] on a rock formation with a scale ranging from "1F" to
"5F" (5 Fuß, 5 feet). It is a question whether this gauge was linked to a large gauge on the castle rock
[RG1]. A similar water gauge which may have been partially preserved is described by the
commissioners at the HS in Pirna.
The second hunger stone [HS2] was supposed to be upstream of the ferry on the right bank. There was
a minimum mark of 1800 situated 4.5″ (approx. 11 cm) above the minimum of 1842. In 1850 the
commissioners stated that the first [HS1] and the third stone [HS1] remained in place, while the
second stone [HS2] was already unavailable at the time of the second commission's work. It is stated
that the reason was the construction of the railway (Protokoll, 1850). Since the railway was on the left
bank, we tend to consider a possibility that the stone disappeared during terrain works for the
construction of a new bridge (opened only later in 1851). The railway was built between1847-1848
and the operation started in 1851.



The third stone [HS3] was located by the commission on the left bank and it still exists. This object is
in centre of our focus. The commissioners described the 1616, 1746, and 1790 marks which were
documented many times later in 1892, 1904, 1911 etc., and also mentioned the 1835 mark (not found).
Unfortunately, they only determined the difference of 5″ (12-13 cm) between the higher minimum of
1616 and the then lowest minimum of 1842 (difference of 11 cm was determined in 2015).
The stone was (see the metodology) divided into four height ranges and the following sides:
embankment side [ES], left side [LS], right side [RS], platform [P] and the highest part of the stone's
ridge [R] (Tab. 3, Fig. 4)

Tab. 3 Division of HS3 stone and list of marks by ranges

| Ranges of water level | ES (embankment side) | R (ridge) | RS (right side) | LS (left side) | P (platform) |
|---|---|---|---|---|---|
| A) 151-175 cm | | 1963 | — | — | |
| B) 111-150 cm | 1536,1616, 1746, 1790, 1800, 1811, 1842, 1868 | — | — | — | 1707, 1842, 1904, 1892, 1893, 1957, 1990, 2003 |
| C) 91-110 cm | 1921, 1934 | — | 1911, 1921 | — | |
| D) 71-90 cm | — | — | 1930,1934,1947 | 1947 | |


The platform P, the ridge part R and the ES side of the stone are about 360 cm wide, the distance
between the bank and the river is about 400 cm. The oldest marks: 1616, 1746, 1790, 1800, 1811,
1842 and 1868 were placed on a side [ES] facing the river bank in the range of 111 to 150 cm. Only
the mark of 1707 was placed on the platform [P] where markings from 1892 to 1904 continued. The
minimum marks 1904, 1911 were simultaneously placed on the right side of the stone [RS]
(downstream). The lack of space also apparently led to rewriting of the inscriptions at the 1911 mark
and a large inscription: *"Wenn du mich siehst …"*. The marking of 1921 returned to the right side [RS]
which was not large enough for a new lower marking below 100 cm. Deeper marks 1930, 1934, and
1947 were placed again on the side of the stone [RS]. The demanding 1947 mark is also on the left
corner [LS] of the stone. The latest markings of 1957, 1990 and 2003 are again on the lower part of the
platform [P] and the mark 1963 on the ridge [R]. Marks of 2015 and 2018 were not placed on the
stone. Overview of water level minima of measured and derived heights is given in Tab. 4. The list of
marks in Table 4 is chronological so that the information is combined into a logical complex (detailed
information is included in the supplement).
Using the example of measurements in 1850, it is possible to clarify the system of rock gauges [RG1],
[RG2] and [OG] linked to hunger stones, and the newly measured heights of the flood of 1784 (2004)
and the minimum of 1842 (2015). An administrator at the Děčín estate, forester and contributor of the
Patriotic Economic Society Seidel (Neue Schriften, 1843), determined the height of the flood mark
1784 on the rock gauge [RG1] as 32′1″10‴ (i.e. 10.16 m) above the minimum stage of 1842 (the height
today is 131.296 m a. s. l. of the Baltic system after equilibration — Bpv). This height after deduction
(i.e. 121.133 m a. s. l. Bpv) is 25 cm lower than mark 1842 on the stone [HS3].






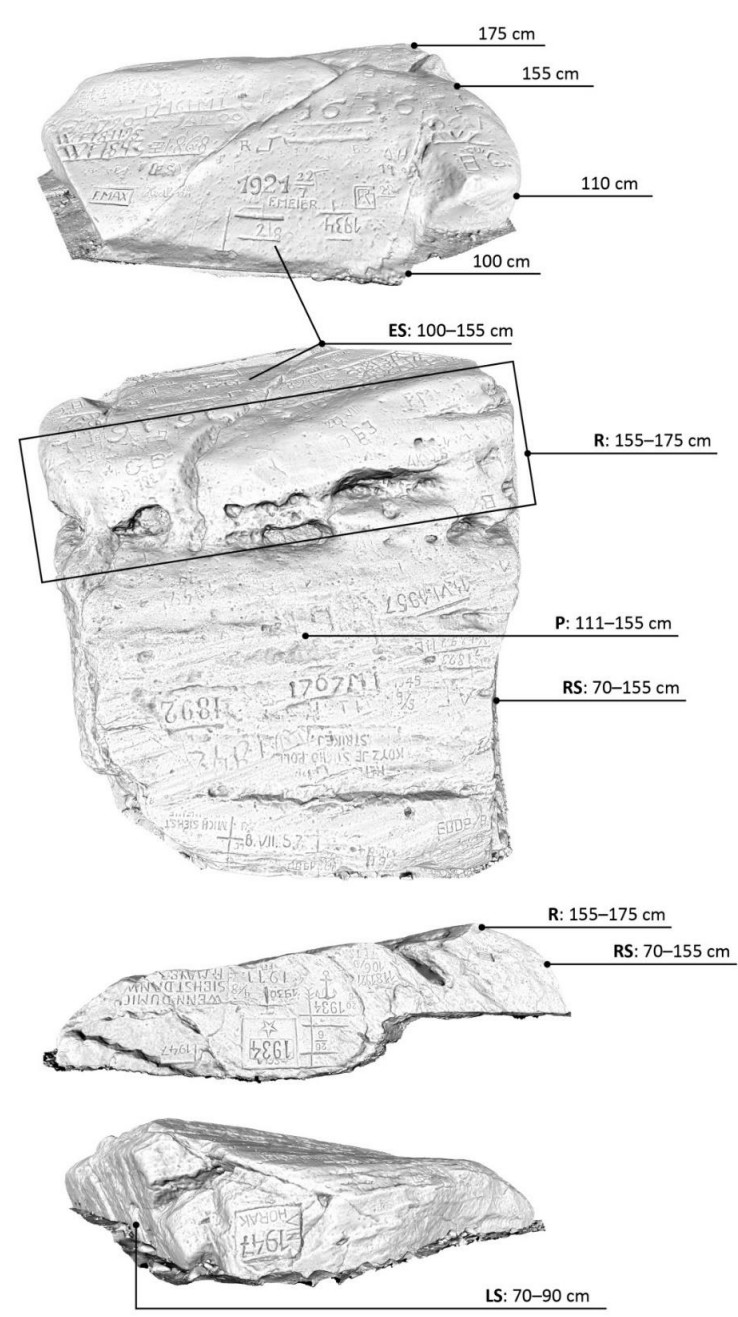

Fig.4 The hungers stone was divided into four height ranges (Tab. 4) and the following sides (from up to down): embankment side [ES], the highest part of the stone's ridge [R], platform [P], right side [RS] and the left side [LS].



Tab. 4 Overview of annual water level minima on hunger stones in Děčín

| Rok | H D | HS2, $H_{1842}$ | | HS1, $H_{1842}$ | | HS3, $H_{1842}$ | | H | $H_{1842}$ | Time | m a. s. l. | Position |
|---|---|---|---|---|---|---|---|---|---|---|---|---|
| | | [´´] | [cm] | [´´] | [cm] | [´´] | [cm] | [cm] | [cm] | | [m] | |
| 1516 | | — | — | — | — | — | — | *119* | **-13** | — | 121.25 | *DZ* |
| 1517 | | — | — | — | — | — | — | *119* | **-13** | — | 121.25 | *DZ* |
| 1536 | **B** | — | — | — | — | — | — | **138** | **6** | — | 121.44 | **ES** |
| 1616 | **B** | — | — | — | — | **5** | **13** | **143** | **11** | *VII, ??, [2?]* | 121.49 | **ES** |
| 1616 | **B** | — | — | — | — | — | — | **137** | **5** | *VII, ??, [2?]* | 121.43 | **ES** |
| 1706 | | — | — | — | — | — | — | *132* | **0** | — | 121.38 | *DZ* |
| 1707 | | — | — | — | — | — | — | **139** | **6** | *VIII/IX?* | 121.45 | **ES** |
| 1719 | | — | — | **8.5** | **22.1** | — | — | **154** | *22.1* | — | 121.6 | *HS1* |
| 1746 | **B** | — | — | — | — | **ND** | — | **150** | **17** | *VIII/IX?* | 121.56 | **ES** |
| 1766 | | — | — | **4.5** | **11.7** | — | — | *144* | *11.7* | *ca 10.12.?* | 121.5 | *HS1* |
| 1782 | | — | — | **8.5** | **22.1** | — | — | *154* | *22.1* | *ca 15.9.?* | 121.6 | *HS1* |
| 1790 | **B** | — | — | **6.5** | **16.9** | **ND** | — | **145** | **12** | *ca 15.8.?* | 121.51 | **ES** |
| 1800 | | **4,5** | **11,7** | — | — | — | — | **142** | **10** | *ca 18. 8. ?* | 121.48 | **ES** |
| 1811 | **B** | — | — | — | — | — | — | **139** | **6** | **9. 8.** **(-1)** | 121.45 | **ES** |
| 1834 | **B** | — | — | **7** | **18.4** | — | — | *150* | *18.,4* | *12. 8.?* | 121.56 | *HS1* |
| 1835 | | — | — | **5.5** | **14.3** | **ND** | — | *146* | *12* | *ca 8. 9.?* | 121.52 | *DZ* |
| **1842** | **B** | **0** | **0** | **0** | **0** | **0** | **0** | **132** | **0** | *ca 25. 8.?,* [2] | 121.38 | **ES, P** |
| **1868** | **B** | | | | | | 1868 | **133** | **1** | *ca 26.8. ?* | 121.39 | **ES** |
| **1874** | | | | | | | 1874 | *128* | **-4** | *ca 1.12. ?* | 121.34 | **T** |
| **1892** | | | | | | | 1892 | **137** | **5** | **28. 8.** **(-5),** [2] | 121.43 | **P** |
| **1893** | | | | ES | | | 1893 | **135** | **3** | **16.7.** | 121.41 | **P** |
| **1904** | **B** | | | ES | | | 1904 | **112** | **-21** | **23. 8.** **(-15),** [5] | 121.18 | **P** |
| **1911** | **B** | | | R | LS | | 1911 | **105** | **-27** | **15. 8.** **(-7),** [2] | 121.11 | **LS** |
| **1921** | **B** | RS | | P | | | 1921 | **104** | **-29** | **2. 8.** **(-9) **,** [6] | 121.1 | **ES, LS** |
| **1930** | | | | P | | | 1930 | **101** | **-32** | *10.9. (+2)* | 121.07 | **RS** |
| **1934** | | | | | | | 1934 | **73** | **-50** | *23.6. (0),* [3] | 120.79 | **RS** |
| **1945** | | | | | | | 1945 | **134*** | **+2** | **9. 5.** **(—)** | 121.4 | **P** |
| **1947** | **B** | | | | | | 1947 | **68** | **-64** | *23.8. (0),* [2] | 120.74 | **LS, RS** |
| **1957** | **B** | | | | | | 1957 | **110** | **-22** | **8.7.** **(0),** [2] | 121.16 | **P** |
| **1990** | **B** | | | | | | 1990 | **110** | **-22** | *2.9. (0)* | 121.16 | **P** |
| **1963** | | | | | | | 1963 | **175*** | **+43** | **(—)** | 121.81 | **R** |
| **2003** | **B** | | | | | | 2003 | **111** | **-21** | **111** | 121.17 | **P** |
| **2015** | | | | | | | 2015 | **86** | **-46** | **86** | 120.92 | **—** |

*HS1 HS2, HS3 Hunger stones in Děčín,* **T** *HS in Těchlovice;* **DZ** *HS in Dolní Žleb;* $H_{1842}$ *water level relative to*
*the height of the mark of 1842 (levels below this mark are in red);* **H** *water level relative to the current Děčín*
*water gauge zero point (120.06 m a. s. l.); * neither annual minimum (AM) nor local minimum (LM) but an*
*indication of contemporary water stage; **the exact AM is denoted by date of 11[th] August and contemporary*
*water level (a value) without any mark;* **Time,** *date of marked minimum,* **thick underlining** *signifies exact day*
*engraved in the stone, probable timing of the mark creation: (-) days before the annual minimum water stage (+)*
*after the annual minimum, ? uncertain value, ?? very rough estimation; [n] n is the total number of marks in a*
*year; italic water level values are derived from another object, timing estimated from another gauge;* **Position**
*placement of DM on ES, RS, LS, P and R sides (Tab. 4), for derived data, the original objects DZ or HS1 are*
*highlighted,* **ND** *— mark registered but not surveyed (Protokoll, 1850)*





4.2.5.Hunger stone in Dolní Žleb (Niedergrund)
In the river map (Elbekarte, 1848), a total of 7 to 8 stones are marked on the right bank of the Elbe
River upstream of Dolní Žleb, followed by another 6 downstream, as indicated in the Protokoll (1842).
At the former customs house (left bank), the Elbe river flow was narrowed by two rock outcrops: the
Monk's Stone (Mönchstein) and the Mill Stone (Malstein) which were removed in 1858 (Protokoll,
1858). Not far from them, in the middle of the stream opposite the church, two stones were identified
in 1842 with the year ending by the figure "16" which was 12″ under water (30 cm). Some sources
(Neue Schriften, 1843) date the marking back to 1516 or 1517. The Commission measured the depths
of the 1616, 1706 and 1842 minima (tab. 5), further depth data were designed to be surveyed
accurately by geodetic levelling and then the stones would have been blasted off as an obstacle. The
regional literature (Focke, 1879; Pažourek, 1998) states that an inscription "I A B R O 1516 — CB
1615 — VC 1634" should have been on the stone which meant *"Ich Andreas Beutel, Richter der Ortes*
*1516", "Christof Beutel 1615"* and *"Christof Vogel 1634"*. According to the latest field surveys (tab.
6), a total of 11 hunger stones were found at the position of 730.55 to 732.01 km, one of them having
the year marking of 1842 (Randák, 2015, 2017a). Identification with the described stones is not yet
possible.
Tab. 5 Marks on a hunger stone in Dolní Žleb surveyed in 1842 (Protokoll, 1842)

| years | $H_{1842}$ | | $H_{DE}$ |
| | ″ | [cm] | [cm] |
| --- | --- | --- | --- |
| **1516** | -5 | -13 | 119 |
| **1517** | -5 | -13 | 119* |
| **1615** | ND | — | — |
| **1616** | 2 | 5.2 | 137.2 |
| **1634** | ND | — | — |
| **1706** | 0 | 0 | 132 |
| **1842** | 0 | 0 | 132 |

*$H_{1842}$ DM water level relative to the level of DM1842, $H_{DE}$ water level relative to the current Děčín water gauge,*
*\*report only (Neue Schriften, 1845), ND — mark registered but not surveyed (Protokoll, 1850).*
Tab. 6 Hunger stones detected by Randák (2015, 2017a)

| No. | Km | Description |
| --- | --- | --- |
| 1 | 730.550 | 1904 (15. 8.) |
| 2 | 730.780 | 1892, "E. Dittrich" |
| 3 | 730.82 | 1892, "Ed. Ditr." |
| 4 | 730.830 | 1892, 1893 "E. H.", "E. D.", heart motif |
| 5 | 730.910 | 1921 ("F. H." ?) |
| 6 | 731.160 | 1892 "F. Hobe" or "Hoke" ? |
| 7 | 731.260 | 1842,1868, 1892, 1904, "V. Witr" ?, "V. Hobe" |
| 8 | 731.180 | "HF" 1892, 1935 |
| 9 | 731.415 | 2015 "13.8." |
| 10 | 731.420 | 1904* |
| 11 | 731.01 | 1904* (at the house of the former ferryman H. Strasser) |

*\* under water at the time of exploration, **?** the inscription is unclear, **in grey** old DMs originated before 1892.*

4.2.6.Hunger stones in Hřensko
None of the Commissions (1842, 1850 and 1858) identified a stone with a year indication. The survey
carried out by experts of the Elbe River Board on 26 August 2107 (flow rate 75 m³·s⁻¹) determined 14
objects with markings, all of which originated after the Commission in 1842 (tab.7).





Tab. 7 Hunger stones detected by Randák (2015)

| No. | Description of the hunger stone |
|---|---|
| 1 | 1928?, "5. 8. (19)? 28" |
| 2 | 1950, „Kladno 1950" |
| 3 | **1874, „K.R. 10/9. 1874"** |
| 4 | 1904, „H. Rausch1904" |
| 5 | ?, „W.W F.D.N" |
| 6 | 1911, 1919, "1911", "1919","3. 8. 1911 ER WK PP" |
| 7 | 1911, "FC 1911" |
| 8 | 1892, "1892" |
| 9 | 1934 "1934" |
| 10 | 1928,1950,"1928"," 1950", "GW" |
| 11 | 1927, "N 1927" |
| 12 | 1927, "1927" |
| 13 | "1928", "1855", *many other inscriptions below the water level* |
| 14 | "1904/ 22.7", "1934", *many other inscriptions below the water level* |

*? the inscription is unclear, **in grey** old DMs originated before 1892.*

4.2.7.Hunger stone in Schmilka
On the right side upstream of Schmilka, the Commission (Protokoll, 1842) found a large stone with
the 1842 mark (4. 9.) which was 4″ (10 cm) below the then current water level. Further, the 1811 mark
was found that was placed higher by 3″ (7.5 cm).

4.2.8. Hunger stone in Stadt Wehlen - Pötzscha
A mark of 1868 remains there until today.

4.2.9.Hunger stones in Königstein
The Commission did not mention any remarkable stone there in 1842, 1850 and 1858. Yet, German
sources mention the year 1681, on another stone 1797, 1914, 1865, 1900, 1911 and 1914
(https://www.umwelt.sachsen.de/umwelt/wasser/download/Dokument_Hungersteine_und_Untiefen.pd
f). In the locality opposite the Prossen village there is a stone that is most often mentioned. Today
there are readable inscriptions with dates of 1868 (20. 9.), 1928 (20. 7.), 1947 (20. 7.), 1963 (31. 7.),
and 2003 (17. 7.). The lowest mark of them relates to 1868 with a correctly marked minimum (in
Děčín the minimum was on 19[th] September). The year 1947 was marked prematurely, so that can
explain that the mark is the highest (in Děčín the difference between 20[th] July and a minimum on 11[th]
August is even 44 cm!). The year 1928 is marked quite correctly, although it is not an annual
minimum (4. 8.) but the difference is very small. On another stone there are newer data of 1963, 2003,
and 2015.
4.2.9.Hunger stone in Pillnitz
None of the Commissions (1842, 1850 and 1858) found there any remarkable stone. However, the
Pillnitz site is, next to Dresden and Meissen, the place of important flood level observations as early as
of 1736 (Pötzsch, 1784). There is a clear inscription of 1778 which is probably not the minimum water
level (see discussion). Marked DM minima: 1893, 1904, 2003, and 2018).





### 4.2.10. Hunger stones in Pirna


It was located near a small gate at the navigation control point but the situation does not exist today.
Nearby, there was a transverse dam opposite to which a flat stone was to be with engraved marks.
According to the Protokoll (1842), the marks of 1616, 1706, 1707, 1746, 1834 and 1835 were
registered and surveyed (the other marks were illegible). Water level at that time was 6″ (0.13 m)
above the inscription *"Waserbau Direction 1842"*, **(Fig.8)**. At the navigation office there was a water
gauge placed on the retaining wall for low water levels (up to 4 Saxon ell units) continuing on the
building (the higher part). The minimum of 1842 was at the level of -1 ell 22.5″ (-1.08 m) below the
zero point. Water level during the measurement in 1842 (on 8[th] September) was at a height of -1 ell
16.5″ (-0.95 m). The difference between the marks of 1616 and 1842 was 5 ″ as in Děčín. In 1850 (on
27[th] September), the water level of -1 ell (-0.57 m) was registered. The measurement was carried out at
that time at water level height 0.38 cm higher than in 1842. The previously described marks were up to
51 cm below the water level. Therefore, there is no reference to hunger stone here. In 1874 (at a time
of catastrophic drought), a new water gauge with a zero point at 110.94 m a. s. l. was set up; if the zero
point of the original water gauge was the same, the minimum in 1842 was at 109,856 m a. s. l.
According to photographs of the current state, the inscription from 1842 and the marks 1707 and 1790
were preserved, the marks of 1616 and 1746 were not found. In addition, they are readable marks of
1782 and 1811. After 1842, the marks of 1859, 1863, 1868, 1873, and 1892 were added. The newer
markings (1904, 1947, and 1952) are probably lower with regard to later channel dredging while the
marks of 1963 and 2003 are higher after the Vltava cascade was opened. On the stone there are 5
scales for particular years like 1707, 1904, 1911, 1842, and 1952 showing more minima in a year. In
2018, the stone was documented by SLUG Dresden experts and the results were presented at a
seminar on flood marks and minimum water levels in Jena in March 2019. There was an exchange of
information between CHMI and SLUG. We provided a sketch of the stone in Pirna which was used to
reconstruct the engraved signs that are exhibited today in the SLUG building in Dresden.
(https://www.thueringen.de/th8/tlug/presse_und_service/veranstaltungsmaterial/2019/01/index.aspx).
The second, newer stone in Pirna has the mark of 1904.
Tab. 8 Marks on a hunger stone in Pirna surveyed in 1842 (Protokoll, 1842)

| years | $H_{1842}$ ″ | $H_{1842}$ [cm] | $H_{DE}$ [cm] |
|---|---|---|---|
| 1616 | 5 | 13 | 145 |
| 1706 | 11 | 28.6 | 161 |
| 1707 | 9 | 23.4 | 155,4 |
| 1746 | 10 | 26 | 159 |
| 1834 | 9 | 23.4 | 155,4 |
| 1835 | 9 | 23.4 | 155,4 |
| 1842 | 0 | 0 | 132 |

*$H_{1842}$ DM water level relative to the level of DM1842, $H_{DE}$ water level accommodation to present Děčín gauge*

### 4.2.11. Hunger stones in Dresden


None of the Commissions (1842, 1850 and 1858) mentioned any remarkable stone. Nevertheless,
pictures of hunger stones are published of the Kotta locality with a year inscription of 1630 (it is
possible that it rather concerns 1636). We have no views regarding the credibility or existence of these
stones. In the Radebeul locality, there is probably a millstone with a year inscription of 1911. In the
Laubegast locality, there are stones with year inscriptions of 1892, 1893, 2003, and 2013. In the
Tolkewitz locality, there is a stone with the 2016 mark. In the Augustbrucke cross-section a low water
level of 1705 was indicated (Pötzsch, 1874), and now, there is also the mark of 2018.

### 4.2.12. Hunger stone in Meissen


We learn about the hunger stone from older literature of the 18[th] century. None of the Commissions
(1842, 1850 and 1858) found any remarkable stone. The only report on the flood marks is conveyed in





literature. Ursinus (1790) mentions in dry 1746 year (see the tab. 4, 9,10) the discovery of various
stones in the Elbe River and one with year markings, pointing out the year 1654.

### 4.2.12. Hunger stone in Strehla

The Protokoll (1842) describes a hunger stone (rock rising from the river) on the right bank of the Elbe
with minima from 1718, 1746, 1790, 1800, 1834, and 1835. The height of 1800 was 5″ below the then
current water stage. The water level at the Strehla water gauge in 1842 was -1 ell 15″ (-0.91 m), in
Riesa, the water level was -2 ells (-1.132 m), and in 1850 only -6″ (-0.14 m) (Protokoll, 1842 and
1850). This stone was probably removed, while another rock block called Nixstein remained there (at
the left bank) formerly dreaded by boatmen, where a depth of 1.60 m was measured in 1850. A
somewhat problematically placed mark was made here in 2018 (https://www.saechsische.de/eine-
hungermarke-fuer-den-nixstein-4001437.html).

### 4.2.14.Hunger stone in Schönbeck near Magdeburg

On 29th May 1858 the Committee recorded the water level at 4′5″ (139 cm in accordance with the
1827-1888 Magdeburg series indicating the water stage at 141 cm). A board with the inscription
marking 29th August 1904 was removed from the river bank and placed in the museum.

### 4.2.15. Notes on creating and specific details of the marks of water minima

There are always fewer records of low water levels (if any) than marks of high water stages, the only
exception is possibly the sandstone Elbe valley between Děčín and Pirna. It is more difficult to make a
mark of the minimum water level than making the flood mark.

(1) It is and it has always been difficult to estimate the correct instant of reaching the minimum level.
More demanding inscriptions were probably made in advance; the designated place was probably
enclosed beforehand by a small barrier so that the mark could be completed at a time when it was
clear that the minimum was reached, i.e. when the water was rising. Therefore, the logical
moment of making the minimum mark is after the minimum has subsided (in the reality 1-15 days
before annual minimum level were this DM levels engraved see tab. 4). However, it is not clear
whether this was a local or annual minimum.

(2) In some years, the level fell again lower, the exact date is given, or a range of water levels for a
given year is made, such as in Děčín for 1904, 1921, 1930, 1934, and 1957. A surprise can be
evoked also by marking the year 1707 in Pirna, as it concerns probably other year. The mark of
1842 seems to have a different meaning being the actual water stage in feet.

(3) The minimum markings are often made upside down (made from the upper side of the stone),
some were made while standing in water or at a lowered position (oriented normally). Therefore,
the engraved lines in such cases are not below the date (in the graphic sense) but above it, thus
closer to the water surface (in Děčín for instance DMs: 1536, 1707, 1892, 1893, 1904, 1911 and
1934).

(4) The marks are completed by monograms (see Pažourek, 1998). The oldest mark from 1616 was
completed by initials F. L., from 1707 by initials M. L. R., and from 1746 by H. M. L., so there is
a possibility that they concern members of one family. Later, in 1790, there are the initials of H.
G. T., in 1800 A.I., in 1811 and 1842 W. E., and the designation is missing for 1821. Another
change is the first year corresponding to the instrumental series, so in 1868 the initials are F. H.,
however in 1892 and re-listed in 1893, the designation contains the initials U. E. The originator of
other marks is probably popular Franz Mayer who is the author of 1904, 1911, 1921 and perhaps
even 1930 markings. In connection with the 1904 mark, the popular inscription *"Wenn du mich
siehst dann weine"* was created. The last mark until the relocation of the original German
population comes from 1934. The originator of the first postwar mark is Mr. Horák. It is therefore
evident that signs of low water levels were accompanied by specific habits.



(5) There are overlapping inscriptions. In view of the place of origin and various perhaps personal,
local, national and even commercial considerations, there were exceptional cases of overlapping
inscriptions. Thus, the 1904 mark, perhaps made by a certain Rotsch, was obscured by the second
inscription: *"Wenn du mich siehst dann weine, Fr. Mayer"* relating to 1911.

**4.3. Assessment of identified water level minima of 1516-2018**

4.3.1. Decade frequencies of 1500-1800

There are no direct water level observations for comparison purposes in the 1516-1726 period.
According to Brázdil et al. (2013, 2015), the 1511-1520, 1531-1540, and 1631-1640 periods had a
higher decade frequency (n = 6 per decade) of drought reports. The coincidence of very low-lying DM
marks (H = 110-140 cm) in 1516, 1517, 1536, 1616, and 1636 with these three decades is evident from
Fig. 5 Brázdil et al. (2015) selected several periods of intense drought for detailed processing on the
basis of an analysis of documentary sources. The years 1534, 1536, 1540, 1590, and 1616 were
selected as extreme cases. In two cases (1536, 1616) there are documented DM marks, but three are
missing. Even so, we can consider our documentation a good match. This result supports the
credibility of the 1516 and 1517 marks which have not been preserved or not yet documented, which
we only know from the Neue Schriften (1845) and the report by Focke (1879). On the other hand,
from 1536 to 1616, no record of water level minima exists in the set for 80 years, although minima of
both extremes in 1540 and 1590 could be expected. From this period we can mention only the height
of 1541 from the Rhine basin. However, in the 1560-1600 period, a very high frequency of floods is
documented, with a recurrence period of 10 years or more ($\geq Q_{10}$) (Elleder, 2015). Although the dry
period does not exclude significant floods at all, in this case it concerned more frequent cases of floods
of approx. $Q_{20}$. We can consider it a period with an average drought occurrence, where according to
monthly rainfall indices at least the index -2 (very dry month) occurred in two or more consecutive
months in 1555 (3 months), 1561 (2), 1562 (2), 1571 (3), 1581 (2), 1589 (2), and 1590 (4). The index -
3 (extremely dry month) occurred only once in 1569 (May) and in dry year 1590 for two months (July
and August) (Brázdil et al. 2013, 2015).

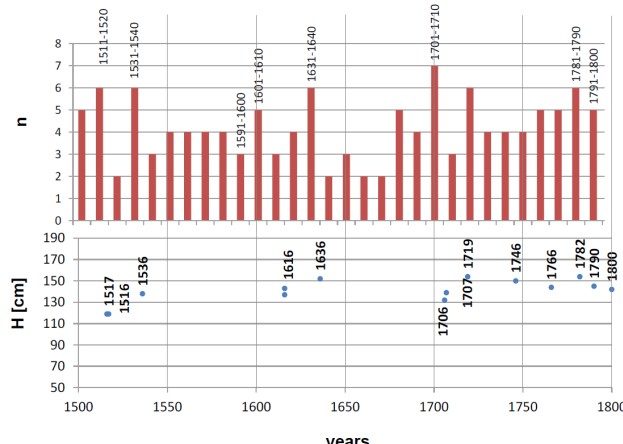


*Fig. 5 Verification of marks of 1500-1800 according to the decade frequency of drought reports by Brázdil et al.*
*(2013), **n** decadal frequencies of droughts, **H[cm]** water level of DM.*

From 1636 to 1707, i.e. for 70 years, there are no marks of minimum water levels. Brázdil et al. (2013)
pointed out that the three decades 1641-1650, 1661-1670, and 1671-1680 had a minimum decade-
occurrence of drought reports (2 cases per decade). Moreover, it is a period of Maunder Minimum
(Eddy, 1976), i.e. the 1640-1720 period, probably the most intensive period of LIA.






4.3.2. The Magdeburg series minima of 1726-1880
Since 1726, we can identify the minima in the years highlighted in Fig. 6 with the help of the
Magdeburg series. A very good time coincidence is apparent for 1746, 1766, 1782, 1790, 1800, 1811,
1835, 1842, 1858, and 1874. The year 1868 is missing, not representing a deviating minimum in
Magdeburg that is more significant later in 1869. The year 1766 represents the only significant winter
minimum which was marked on hunger stones. On the contrary, winter minima of 1818, 1823 and
1862 are missing.

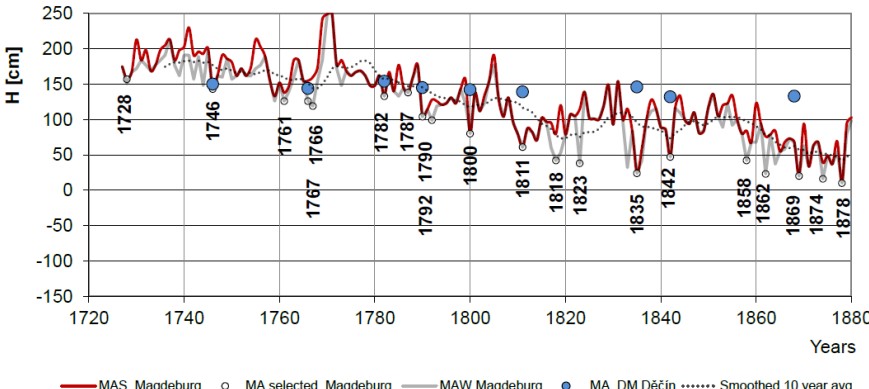


*Fig. 6 Verification of marks in the period of 1726-1800 according to the annual (grey) and summer (red line)*
*minima of the Magdeburg 1726-1800 series with annual minima identified (and derived) from the marks on the*
*HS3 hunger stone in Děčín (blue circles).*

The water level DM minima are plotted in the water level scale of the current water gauge in Děčín. A
coincidence regarding the water level (1746) is completely random (Fig. 6). However, there is a
noticeable difference in the trend of annual lows of both series. We also emphasized the effect of
overall minima, so the graph also separates the winter minima which show the downward trend, for
example, just before 1746.
It is worth noting that the winter minimum of 1823 is not shown on the Elbe HSs but in view of timing
it corresponds to the low water levels of the Rhine. The only significant summer minima that are not
documented on the HSs in the Czech part of the Elbe are around 1760, 1858 and 1878 (see
Discussion).
4.3.3. The Děčín series minima of 1851-2018
If we compare the results with the Děčín series, i.e. with direct measurements in the vicinity of the
HS3 hunger stone, the deviations of the marked and measured annual values are minimal. Until 1957,
there are 11 year lows (not counting local minima) which we can evaluate and 8 of them have a
deviation lower to 4 cm. The result worse than 5 cm is detected for the marks 1911 (+7), 1921 (+9),
1930 (+5), 1947 (-6 cm), and 1957 (+5) (see the graph in Fig. 7). In 1921, the local minimum was
correctly marked; the annual minimum was not marked. Minima marked later, in 1963, 1981 (missing
in the figure), 1990 and 2003 are not as important as the older extremes. In their originating, modern
anthropogenic influences and partly misunderstanding of older traditions are manifested. This also
applies to the prematurely made mark in Těchlovice. The 2003 mark is made well.





In conclusion, we can state a good match of the minima detected which, moreover, are mostly
representative of the largest extremes. However, this is not true entirely, as some years such as 1540,
1590 or 1761 are missing. This is a great motivation for the next stage of future work.

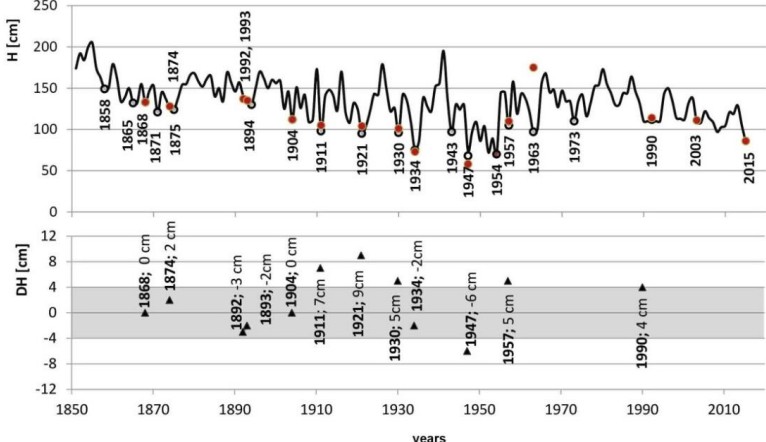


*Fig. 7 Coincidence of annual water level minima at the Děčín station and altitudes measured on the HS and HS3*
*hunger stones in Děčín and Dolní Žleb. **H** is water level. Deviations **DH** are highlighted in the lower part of the*
*graph. In grey are highlighted the precisely marked height (HP) at a deviation of 0-4cm, outstanding are*
*approximately marked height (CP) at a deviation of 4-8 cm and more.*

**5. Discussion**
**5.1. Credibility of minimum flow marks, uncertainty at some of them and certainty of**
**untrustworthy of other**

There is no need to doubt the credibility of the low water level marks in Děčín from 1868 to 1957.
When interpreting them, however, it is necessary to know the described changes, whether they are
changes in the channel or flow rate enhancement by the Vltava cascade. These are annual or local
minima marked with the greatest possible care. It is also obvious that older marks in the 19[th] and 18[th]
centuries were made in the same way and with the same intentions. Can this claim be extended to the
past, i.e. to the 17[th] and 16[th] centuries, and is this finding valid for other hunger stones both in Bohemia
and Saxony?
It would probably be appropriate to prove the connection of heights in Děčín, Dolní Žleb, Schmilka
and Pirna. However, when verifying the relationship between Pirna and Děčín, we can compare only 4
concurrent records. These are the years 1616, 1707, and 1842. Since we use the relative difference to
the water stage in 1842, we can only compare the three remaining heights of 1616, 1707, and 1746.
The relationships of 1616, 1707, and 1842 are linear, somewhat different is the water stage in 1746
where the difference from the expected value is greater than 10 cm. Perhaps only a local minimum
(LM not AM) was marked in Pirna. However, we only use the published data from 1842 and from
1843 and it is not entirely certain that the commissioners found and surveyed the lowest mark for a
given year. Verification is still difficult; we do not see this mark on the current stone in Pirna-
Oberposta.





We can recommend further field survey in the future (next one especially in Dolní Žleb), levelling and
scanning of other objects, especially the stone in Pirna. For detailed analysis and search for relics of
older marks, it is not possible to rely solely on photographic documentation. Comparative older
photographic material (Fig.8) and detailed inspection of scanned 3D objects is required.

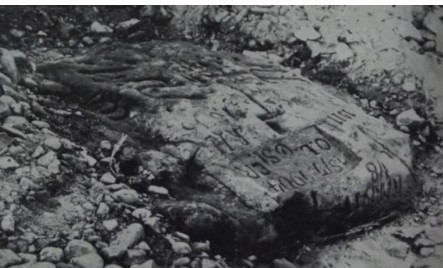


*Obr. 8. Picture from the Český svět magazine, No. 51 from 25th August 1911. It shows a completely unknown*
*hunger stone. The following years are engraved: 1835, 1904, 1911, 1873, and 1(?)76 (1576, 1876 or 1516?).*
*This picture was found recently in the National Museum archive by Zvonimír Dragoun. The locality is unknown*
*and the existence is also unverified.*

Since we can trust DM epigraphic sources, it remains to point out other published sources in 1842-
1843. These are compilations of the then measurements by the commissioners-hydro-technicians and
possibly subsequent processing by the Statistical Office of the Kingdom of Saxony, or the Patriotic
Economic Society of the Czech Kingdom, respectively. They point to other low levels that we
expected and that could not be verified. This is, for example, the height of 1590. A report drawing
certainly from the results of the Commission in 1842 and therefore the Protokoll (1842) appeared in
the Adler Magazine (No. 13 of 13 January 1843). There is water level reported in Dresden as 2 ells 3
inches below zero point and a series of low levels, of which we choose those that could not be
documented in situ or verified in the scientific literature: 1590, 1634, 1635, 1637, 1660, 1666, 1669,
1678, 1681, 1686, 1705, 1716, 1718, 1726, 1761, 1789, and 1794. Another remarkable source is an
article in the Prague summary report Encyklopädische Zeitschrift des Gewerbewesens (3rd edition of
the new series from 1843, Statistik der Gewerbe und Handel, pp. 86-93), which draws on the
Preussisch Staat Zeitung Nr. 354. The same data were published in a more popular way in educational
journals such as "Das Pfennig Magazine für Belerung und Untrehaltung" (1843, 11 March, No 10).
The exact heights published there are given in Tab. 9. The Gewerbe Blatt für Sachsen (No. 5/1843,
https://digital.slub-dresden.de/werkansicht/dlf/69679/1/), a technical magazine, states in the
explanatory note that a minimum mark of 1590 was indistinctly recognized on an unnamed hunger
stone or object in Rathen between HS in Königstein and HS in Stadt Wehlen-Pötzcha. It is not clear
whether the mark was too deep or unreadable and its height was therefore not stated.










Tab. 9 An overview of the Saxon DM-type sources (edition of the new series from 1843, Statistik der
Gewerbe und Handel, pp. 86-93)

| | Saxony $H_{1842}$ | | | Pir $H_{1842}$ | Sch $H_{184}$ | Str $H_{184}$ | $H_{DE}$ |
|---|---|---|---|---|---|---|---|
| Year | ["] | ['''] | [cm] | [cm] | [cm] | [cm] | m. a. s. l. |
| **1615** | *17.5* | | **45.7** | — | — | — | 177.7 |
| **1616** | *3.5* | *0.5* | **9.1** | 13 | — | — | 141.1 |
| **1635** | *9* | | **23.5** | — | — | — | 155.5 |
| **1636** | *8* | | **20.9** | — | — | — | 152.9 |
| **1705** | *11* | | **28.7** | — | — | — | 160.7 |
| **1706** | | | **—** | 28,6 | — | — | |
| **1707** | *4.5* | *0.5* | **11.8** | 23,4, E | — | — | 143.8 |
| **1718** | | | **—** | — | — | ND | |
| **1746** | *10* | | **26.1** | 17 | — | ND | 158.1 |
| **1761** | *5.5* | *0.5* | **14.4** | — | — | — | 146.4 |
| **1782** | *11* | | **28.7** | E | — | — | 160.7 |
| **1789** | *14* | | **36.5** | — | — | — | 168.5 |
| **1790** | *6* | | **15.7** | — | — | ND | 147.7 |
| **1794** | *11* | | **28.7** | — | — | ND | 160.7 |
| **1800** | *8* | | **20.9** | E | — | — | 152.9 |
| **1811** | *6* | | **15.7** | E | 7,5 | — | 147.7 |
| **1811** | *6.5* | *0.5* | **17.0** | E | — | — | 149.0 |
| **1834** | *8* | | **20.9** | 23,4 | — | ND | 152.9 |
| **1835** | *8* | | **20.9** | 23,4 | — | ND | 152.9 |
| **1842** | *0* | | **0** | 0,E | 0 | — | 132 |


*Saxon inches ["] and line units ['''], **Pir** Pirna HS, **Sch** Schmilka HS, **Str** Strehla HS, **$H_{1842}$** DM water level*
*relative to the level of DM1842, E existence is verified, **$H_{DE}$** water level relative to the current Děčín water*
*gauge, **ND** mark registered but not surveyed.*
If we take this source into account, and we combine this data with the data already presented, we get a
slight shift somewhere, but the overall picture and trend confirm information on the minima of water
levels from hunger stones in Bohemia. Another source is the report of the Patriotic Economic Society
(Neue Schriften, 1845) where the forester and observer of the Děčín- Podmokly station gives the exact
height of the marks (Tab 10). It is partly a compilation of the heights from Děčín and Dolní Žleb, the
data are very similar or the same (1616, 1707, 1746, 1811, 1835, and 1842). Differences over 8 cm
show only DM of 1766, minor differences are in the years 1782, 1790, and 1800. However, there are
also data for 1834, 1516, and 1517. To complement the Děčín data, the minima of 1516 and 1517 were
mainly used. We assume that, as a forester and a meteorological observer, A. Seidel could supplement
the report of the commissioners (who had only limited time to survey) from his own examination in
Dolní Žleb and Děčín, where he lived. The years 1516 and especially 1517 were really dry, as
evidenced by contemporary descriptions in the Old Czech Chronicles (SLČ), in particular, describing
rather meteorological and phonological parameters of drought (e.g. harvest already on 29[th] June).









Tab. 10 Compilaton of the Czech DMs (Neue Schriften, 1845)

| Year | H₁₈₄₂ | | | Comparison with objects on the Czech side | |
|------|-------|---|----|------------------------|---------------------------|
| | *Inch* [″] | *Line unit* [‴] | *Cm* | H₁₈₄₂ [cm] | Object HS and (sources) |
| 1516 | -5 | | **-13.1** | -13 | DZ, (NS,P) |
| 1517 | -5 | | **-13.1** | -13 | DZ,(NS) |
| 1616 | 4 | 4 | **11.3** | 11 | **HS3, (L)** |
| 1707 | 3 | 4 | **8.7** | 6 | **HS3, (L)** |
| 1746 | 6 | 6 | **17.0** | 17 | **HS3, (L)** |
| 1766 | 10 | 2 | **26.5** | 11.7 | HS1, (P) |
| 1782 | 6 | 8 | **17.4** | 22.1 | HS1, (P) |
| 1790 | 6 | 6 | **17.0** | 12 | **HS3, (L)** |
| 1800 | 6 | 10 | **17.9** | 10 | **HS3, (L)** |
| 1811 | 3 | 1 | **8.0** | 6 | **HS3, (L)** |
| 1834 | 7 | 0 | **18.3** | 18.3 | HS1, (P) |
| 1835 | 6 | 0 | **15.7** | 14.4 | HS1,(P) |
| 1842 | 0 | 0 | **0.0** | | HS3, (P) |

*Austrian inches [″] and line units [‴], **H₁₈₄₂** DM water level relative to the level of DM1842, **Object HS, HS1, HS2, HS 3** hunger stones in Děčín (tab. 4) or in **DZ** Dolní Žleb (tab. 5), (**X**) sources of data **P** Protokoll (1842), **NS** Neue Schriften (1845) and **L** levelling and surveying in 2015 (tab 4); **in grey** a very good agreement denoted.*

## 5.2. Bad and doubtful markings

In the promotional photographs issued as postcards we can find supposed minima marks that do not correspond to reality (correction in parentheses) such as the years 1745 (1746) and 1858 (1868). The often published postcard with a lady in a hat by E. Rennert (as in Brázdil et al., 2015, 2019a) and an article in the anthology indicate an inscription of 1417 (Pažourek, 1998) in the left part of the plateau at the river. Is it possibly a misinterpretation or a complete forgery? In these places, there is now an inscription of 2003, but there is no indication that there is any mark, not to mention that the date would be necessarily made using Roman numerals. There used to be completely or partially wiped out inscriptions of the minimum of 1904 and the inscription "1904 Weh", or misery or suffering. These inscriptions practically disappeared.

In the river side part of the Pillnitz castle there are signs including a year marking of 1778. By comparison with the mark heights in Magdeburg and the descriptions in the documentary sources it can be considered rather to mark the year of repairing the castle in 1778 or even the anniversary of its founding in 1718. But then it should be correctly marking of 1718.

## 5.3. Probable connections between flood marks and hunger stones in Pirna and Děčín

It is remarkable that we find virtually the same tradition and the same DM marks in Děčín and Pirna, on the Saxon and Czech sides. At that time from the 13th to the beginning of the 15th century, today's Saxon Pirna was part of Bohemia. In 1432 the towns were hit by a catastrophic flood, the height of which is marked in Děčín next to the RG1 rock water gauge. In reverse, in 1515, Děčín became the property of aristocratic families from neighbouring Saxony, first of the Lords of Salhausen and from 1534 of Bünau (Schattkowsky, 2003). Until 1628, i.e. for 94 years, this family was in possession of Děčín and Weesenstein estate in the vicinity of Pirna. At that time, the oldest identified low-level signs of 1536 and 1616 were made on the HS3 stone in Děčín. Solely the literature documented marks of low water levels were made in 1516 or 1517 (Neue Schriften, 1845), i.e. at the time of the Salhausens. With the beginning of the Thirty Years' War (1618-1648) and re-Catholicisation in Bohemia in 1626 Pirna became the centre of Czech exiles. It is evident that Děčín and Pirna are bound by one river,





cruise and partly by common history. It is therefore not surprising that we find an analogy in the area
of documentation of flow minima.

**5.4. Relationship between the Rhine and Elbe minima**

The alluvial-pluvial regime of the Rhine predetermines the seasonality of the Rhine minima which
occur rather in autumn and winter. That is mostly later than at the Elbe where there are mostly summer
minima. The very dry period of 1536 to 1541 is defined particularly by the Elbe and the Rhine minima
(Tab. 11 a, b). Only from the literature the mark of 1654 in Meissen is known, when there are also a
number of reports from the Rhine basin. Almost perfect concurrence is represented by the minima of
1766 and 1767. The very warm and dry period of 1790-1794 was evident in both river basins. The
lows also coincide in 1800 and 1858. In the Rhine basin, drought was more significant. In the Elbe
river basin, the catastrophic flood changed the situation at the turn of July and August, which affected
upper Elbe basin and mainly the Krkonoše and Krušné hory mountain areas (Elleder, 2015).
Tab. 11a Documentation of minimum water levels in the Rhine basin according to Wittman (1859),
and of the Elbe minima on the basis of documented DM marks (1303-1755)

|  | **The Elbe** | **The Rhine** |
|---|---|---|
| **1303** | — | Olten, Strasburg, (W) |
| **1516** | DM, DZ | — |
| **1517** | DM, DZ | — |
| **1521** | — | DM Unkelstein, (BT) |
| **1536** | DM, DE | |
| **1541** | — | DM Laufenstein, (W) |
| **1544** | DM  STA (W) | — |
| **1567** | — | DM Unkelstein, (BT) |
| **1590** | DM RA | — |
| **1615** | DM DE, Sax | — |
| **1616** | DM DE, Sax | — |
| **1627** | DM  Sax | — |
| **1631** | DM  Sax | — |
| **1634** | DM DZ | — |
| **1635** | DM  Sax | — |
| **1636** | DM  Sax | — |
| **1637** | DM  Sax | — |
| **1639** | — | DM Unkelstein, (BT) |
| **1654** | DM ME | Bacharach (Altarstein, 25 people standing on it), (W) |
| **1660** | DM  Sax | — |
| **1666** | DM  STA, Sax | — |
| **1672** | — | Olten, Staad, Konstanz (Horn), (W) |
| **1678** | DM  Sax | — |
| **1681** | DM, KO | — |
| **1686** | DM  Sax | — |
| **1692** | — | DM Laufenstein, (W) |
| **1704** | — | DM St. Goar, (W) |
| **1705** | DM  Sax | — |
| **1706** | DM DE, PI, Sax | — |
| **1707** | DM DE, PI, Sax | — |
| **1718** | DM ST | — |
| **1719** | DM DE | — |
| **1725** | — | DM Mammern, DM Konstanz, (W) |
| **1726** | DM  Sax | — |
| **1746** | DM DE, PI, ST | — |
| **1749** | — | DM Rheinau, (W) |
| **1750** | — | DM Laufenstein ,Kolln – bridge pillars, Bacharach, |
| | | |








Tab. 11b Documentation of minimum water levels in the Rhine basin according to Wittman (1859), and of the
Elbe minima on the basis of documented DM marks (1755-1858)

|  | The Elbe | The Rhine |
|---|---|---|
| **1755** | — | DM Mammelbach, (W) |
| **1761** | DM  Sax, (GBS) |  |
| **1766** | DM DE | — |
| **1767** | — | Kolln, bridge pillars,  (W) |
| **1782** | DM DE, PI, Sax | — |
| **1785** | — | DM Mammebach, (W) |
| **1789** | DM  Sax (NI), (GBS) |  |
| **1790** | DM DE, ST, | — |
| **1792** | — | The lowest stage in Bodensee, Mammern, (W) |
| **1800** | DM DE, SCH, PI, ST | The lowest stage of the Rhine in 30 years, Mainz |
| **1811** | DM, DE, PI | — |
| **1823** | — | Very low water stage of the Rhine |
| **1834** | DM DE, PI, ST | — |
| **1835** | DM DE, PI, ST | — |
| **1842** | DM DE, DZ, PI | — |
| **1848** | — | DM Laufenstein, (W) |
| **1858** | DM Pirna | The lowest stage of the Rhine, (W) |

*DM drought mark, DE Děčín (Tab. 4), DZ Dolní Žleb (Tab.5), SCH Schmilka, KO Konigstein, PI Pirna, ME*
*Meissen, STA Stade, and Sax Saxony (Tab. 9). Other sources in brackets GBS (Gewerbe Blatt für Sachsen NO.*
*5, 1843), W (Wittmann,1859), BT (Börngen, Tetzlaff, 2001).*
A comparison of the duration of the tradition of making minimum markings in the Rhine and Elbe
basins does not clearly indicate a longer tradition in either area. More interesting is a graphical
overview of data from the Czech and Saxon DM sources (Fig. 9). It is apparent that the downward
trend pointed out by reputed geographers and water managers (Burghaus, Grebenau, Wex, Harlacher
and others) in the measured series has been apparent since about 1746, even at lows recorded on
hunger stones. In the case of Děčín, it is clear that during the coldest period of the LIA, the Maunder
Minimum (Eddy, 1976) could have a positive effect on the Elbe runoff, although, for example,
Ogurtsov (2019) illustrates an even deeper minimum in the first half of the 15[th] century.






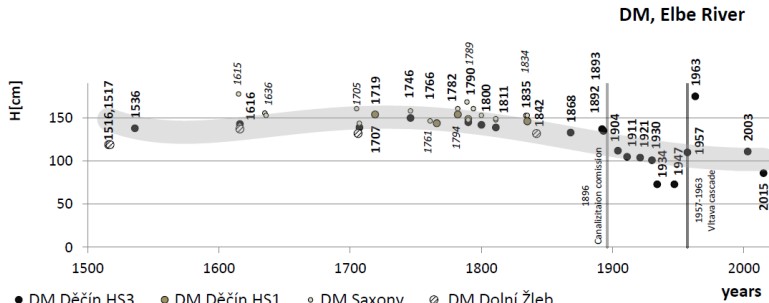



Fig. 9 The graphical overview of DM data from the Czech and Saxon areas (upper part). *The **black and grey circles** represent DM minima derived from the hunger stone in Děčín, **hatched circles relate** to the stone in Dolní Žleb. This ensemble is completed by minima from the Saxon data (small circles), source (tab. 9). **Gray line** highlighted the trend of DM minima. The graphical overview of DM data from the Rhine basin (lower part). Water level minima derived from the Laufenstein DM (**black rectangles**), other epigraphic documentation (**hatched rectangles**) (tab, 11a, b).*

Unfortunately, marks of 1516 and 1517 and their position are known only through the testimony of A. Seidl of Děčín and from an indication in the Protokoll (1842). However, the positions of the 1536 and 1616 marks increase their credibility. The downward trend since 1746 in Děčín cannot be explained only by hypothetical deepening of the profile or as a result of the shortening of the Elbe in the case of Dresden and Magdeburg. The fact that the runoff may have been comparable to the period after 1842 and even lower before the onset of the Maunder Minimum, may be useful knowledge about the status of the basic flow and the status of groundwater. In the case of the Rhine we have very little data available. The existing ones, however, do not contradict previous considerations. Again, there are two important time points, the years 1541 and 1750. Interpretation of other reports on hydrological drought from the Maunder Minimum period is a matter of future studies.

## 6. Conclusion

Hunger stones with low water marks are a phenomenon that has been and is regionally limited to the Upper Rhine basin and the Elbe River. In other regional areas, we have not been able to find an analogous activity where, for centuries, minimum water levels would have been marked. In the Rhine basin, the water level of the Lake Constance and the Rhine level in the area downstream of the confluence with the Aare River to Cologne were marked. While very few of the former objects with low-level marks are available in the Rhine basin, the situation is still favourable in the sandstone part of the Elbe canyon from Děčín to Pirna and its surroundings. There are at least 27 objects on the



Czech side and at least 10 stones on the German side, mainly with signs mostly from the 20[th] century.
Still, several of them are part of an older tradition before 1892 or 1842. Of these, we can only be sure
of the stones in Těchlovice, Děčín, Dolní Žleb, Hřensko, and Pirna. According to the existing findings,
the oldest marks from the 17[th] and 18[th] centuries have been preserved only in Děčín and Pirna, even
though they used to be in several places, and we are not sure about Dolní Žleb. A number of stones in
the navigation route, including the hunger stones, were recommended for blasting by the navigation
committees in 1842 and 1850.
An exceptional situation is in Děčín and Pirna in particular. It consists in the existence of very old
records of minimum water levels and the existence of old records of water levels. In Děčín, moreover,
the 590-year-old flood marks and the 490-year-old low water level marks are combined in one logical
complex. It is evident that the motivation for making the low water marks was related to navigation
conditions in the Elbe canyon. In fact, this tradition was made possible by availability of the local
material, sandstone in the form of rocky outcrops or boulders, into which the marks could easily be
cut, engraved or painted. The minimum signs at the individual objects in Děčín are related to the
dedicated water gauges and markings of the navigation depth which was about 93 cm for half load and
130 cm for full navigability around 1842. The old rock water gauge for high and low water levels and
its projection on the first of the three Děčín stones served the safe loading and passing as good as the
later water gauge in the city.
We have shown that the years with marks or crosses are credible evidence of the occurrence of flow
rate minima, mostly annual minima. If there were other minima in the year, additional lines were
made, forming an occasional water gauge for the given year. Obviously, the originators efforts were to
capture the annual minimum as accurately as possible, and the guarantee of reliability was often their
signature, name or initials. The marks correspond to the measured water levels of the systematic series
and are relatively representative to the important minima of the Magdeburg 1727-1880 and Děčín
1851-2019 series. The correlation of the 1868, 1892, 1893, 1904, 1911, 1921, 1928, 1930, 1934, 1947,
and 1957 markings (DM) in Děčín with the series of measurements shows mostly a match with
differences lower to 4 cm, exceptionally larger. Therefore, we assume the same accuracy, i.e.
compliance with real minima at the same level for marks from the 1516-1867 period.
According to the observed water level minima in the 16[th] and early 17[th] centuries, the minima were at
the same and probably even lower level than 1842. No completely reliable water level minima marks
are yet available for the Maunder Minimum [MM] period in the Czech territory. Marks of 1654
(Meissen) and 1681 (Konigstein) are documented only by more remote literature and their height is
unknown. The exceptions are marks at the end of MM in 1706 and 1707. Levelling measurement of
marks on two stones and creating a 3D model of the Děčín stone by scanning helped to understand the
tradition of water level recording, to rehabilitate the value of marks on hunger stones and to bring new
very reliable data on occurrence of hydrological drought in the historical period.
However, many other questions also emerged from the survey. The question is not whether it makes
sense to document the DM marks, but rather how much of the former collection remained after
regulating the Elbe and operating a chain cruise locally. We are confident that further field and archive
research will bring an opportunity to obtain valuable data on hydrological drought in the past. The
profitability of the resources and time spent on exploration and processing is evident.








**Data availability.** The measurement record and the survey notebook data such as historical records,
Magdeburg, Dresden and Praha, used in the paper, are available from the corresponding authors.
**Competing interests.** The authors declare that they have no conflict of interest.
**Author contributions.** LE prepared the archive and historical sources. LE and LK prepared the field
survey and measurement. TK analysed the object of HS with MeshLab software and JŠ worked with
GIS applications, prepared maps and illustrations. All authors participated in interpretation of the field
data and the results.
**Special issue statement.** This article is part of the special issue "Droughts over centuries: what can
documentary evidence tell us about drought variability, severity and human responses?". It is not
associated with a conference.
**Acknowledgement**
We thank Mr. Zvonimir Dragoun for geodetic surveying of the flood marks in 2004 and 2005 along
the Elbe between Mělník and Děčín and of the low water level marks later in 2015. We relay sincere
thanks for numerous friendly consultations and advice to Mr. O. Kotyza from the Litoměřice Museum,
as well as for great willingness and consultations to the Director of the Museum in Děčín, Mr.
Pažourek and Mrs. H. Slavíčková from regional Archive in Děčín. It is impossible to imagine our
study without the hard work and interest of the team around Mr. Randák from the Elbe River
Administration. The friendly advice and persistent optimism and interest of Mrs. Zlata Šámalová was
a source of valuable information as well as energy. We also thank our colleagues from the SLUG
Dresden. At last but no least we thank for translation Mr. V. Dvořák.

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
