# Peer review of "Low Water Stage Marks on Hunger Stones: Verification for the Elbe River from 1616 to 2015"

_Climate of the Past, 2019_

## Referee Comment (RC1) · Anonymous Referee #1 · 12 Dec 2019

The paper addresses a very interesting issue of historical hydrology, dealing with hydrological droughts based on documented epigraphic records of low flows in central Europe. This type of information is rare, which increases the scientific relevance of this paper. The objectives of this paper are well described, and the methodology provides a step forward to previous manuscripts on this topic, by using laser scanner to provide a 3D record of the hunger stones. The results show a detailed record of cluster periods with low flows in the Elbe River, and relations with some historical climatological periods such as the Minimum Maunder. Particularly on the first part of the paper, it is not very easy to read, and some sentences are too long. English use may need some improvements. I cannot help on this because I am not native speaker, but I have provided some editing comments. In summary, the paper deserves publication after

some moderate/minor changes.

Specific comments:

Lines 97-98. Suggested modification of the sentence. A remaining issue is to verify the credibility of the information on low water levels, and its transformation to provide robust information on runoff previous to 1825 and even before 1726.

Lines 104-106. Suggested change These palaeoflood indicators comprise various type of sedimentary (e.g. slackwater flood deposits) and botanical evidences such as impact marks and damages on trees (Benito et al, 2004, 2015).

Line 106-107. However, similar methods for estimating low water levels and flow rates are seldom addressed, with some exceptions (Shamir et al., 2013). Shamir, E. et al., 2013. Geomorphology-based index for detecting minimal flood stages in arid alluvial streams. Hydrol. Earth Syst. Sci., 17, 1021-1034.

Line 107-108. Therefore, low water level indicators available through documentary sources are unique data records (Brazdil et al., 2018) for recording past hydrological droughts, with the precision given by physical imprints provided by epigraphic marks.

Line 129. The specific issues and questions addressed are: Line 133. Are there consistent relations in the heights of stage minima among different stones?

Line 136. Suggest change the title 2. The Elbe river region in the Czech Republic

Line 145. Delete "rock"

Line 167. Ploucnice River and Jilovsky stream should be placed on map in Figl 1

Line 174. Insert in brackets like this (1 ell=59 cm)

Line 183. (see chapters on methodology and documentary sources).

Line 196. Check if the proper word is acquired or recorded by

Line 361. The brackets are confusing. Probably better as …..in Strahov, Wiesenfeld,

1844).

Line 426. Rhine from 70 AD to 1858 . . . Please, check if 70AD is correct or some numbers are missing.

Line 482. . . . Walter, 1901) of which reported altitudes exist for 1541, . . ..

Line 494. Figure caption is confusing until the text is read. Probably change the caption as follow. Fig. 2. Drawing documenting the position of the hunger stones known as Ara Bakchi, Altarstein or Elfenstein near Bacharach, perhaps in the dry season of 1636, 1639 or 1642 (Merian, 1645).

Line 513. . . . water levels of the Elbe River occurs typically from June to.. Line 571. Add space after between.

Line 649. Perhaps you meant August 2017

Line 666. In the locality opposite to Prossen. . .

Line 678. Marked DM minima includes years 1893..

Line 738. . . . than marking the flood mark, due to the following reasons:

Line 763. Insert in brackets the translation of the inscription, otherwise we cannot follow the meaning of the popular inscription.

Line 916. Perhaps you meant "phenological"

---

## Referee Comment (RC2) · Neil Macdonald (Referee) · 23 Dec 2019

This represents an interesting and detailed analysis of 'hunger stones', used as historical drought markers. The use of 3D laser scanning represents a novel and interesting application, extending beyond previous studies of these markings.

The paper is worthy of publication, however as a native English speaker it requires considerable work to ensure that the paper is articulating the findings clearly and concisely. I have attempted to help with the annotated comments; however, I am unable to work through the whole paper in detail because of time constraints, this requires extensive reworking.

I would recommend reviewing the sub-heading titles and shortening them in places e.g. 5.1 is too long. Figure 4: it would be helpful to have a full image of the stone as an insert depicting the four sections, and then present these four sections as you have them - the combined image is embedded in Table 4.

The paper is a valuable contribution to the literature, but requires extensive reworking, editing and proof reading

Please also note the supplement to this comment:
https://www.clim-past-discuss.net/cp-2019-113/cp-2019-113-RC2-supplement.pdf

[Figure]

**Supplement:**

[revised manuscript text omitted]

---

## Author Comment (AC1) · 16 Mar 2020

Libor Elleder (CA): Answers to Neil Macdonald (NM)

NM The paper is worthy of publication, however as a native English speaker it requires considerable work to ensure that the paper is articulating the findings clearly and concisely. I have attempted to help with the annotated comments; however, I am unable to work through the whole paper in detail because of time constraints, this requires extensive reworking. 1/ I would recommend reviewing the sub-heading titles and shortening them in places e.g. 5.1 is too long.

CA The text was proofreded and by a native spakerm a professional proofreader.

NM Figure 4: it would be helpful to have a full image of the stone as an insert de-

picting the four sections, and then present these four sections as you have them - the combined image is embedded in Table 4.

CA: This is a good idea. Unfortunately, the pictures available are very similar to the upper part of the scan in Fig., 4 however due the confusing shadows and light and bad reading of the marks are not suitable for publication.

Dear reviewer, I hope , I accepted and corrected all points you highlighted.

NM, Line 19

[revised manuscript text omitted]

NM, Lines 82-84

CA: However, if we want to describe how the rainfall deficits and other weather influences were reflected in the runoff from the surveyed river basin, the options we have so far are rather limited.

NM, Lines 87-88

CA With the help of deficit volume analysis with a fixed annual (Q95) and variable monthly threshold (Q95m), Brazdil et al. (2015) identified the drought events of 1868 and 1874 as comparable to the 1904, 1911 and 1947 dry periods. NM, Lines 90-94

CA The authors elaborated in detail the selected dry years of 1808, 1809, 1811, 1826, 1834, 1842, 1863, 1868, 1904, 1911, 1921, 1934, 1947, 1953, 1959 and 2003, i.e. 8 cases in each century representing a total of 16 cases selected on the basis of the lowest Z-index and SPI1 values out of 10 homogenised precipitation series (Brázdil et al., 2012). The evaluation of particular years includes the meteorological and synoptic conditions, drought impacts, monthly values of air temperature, precipitation, SPI1, SPEI1 and Z-index. In the identification of hydrological drought in the 1860s and 1870s, a similar result was reached by Elleder et al. (2019) when analysing the catastrophically dry year 1874 by analysing the newly reconstructed series of water levels in Prague (1825-1890).

Remark: In this study was used standardized precipitation SPI-1 index for month to evaluate yearly distribution of precipitation in selected years

NM, Lines 97-101

CA: I included two of citations: Wilhelm et al (2019) and Schulte et al (2019)

But what credible documents of low water levels existed before 1851 (the start of record-keeping in Děčín), 1825 (the start of record-keeping in Prague) or 1727 (the start of record-keeping in Magdeburg)? Based on reconstructed data on temperatures and precipitation between 1766 and 2015, Hanel et al. (2018) indicated extreme deficits in precipitation, runoff and water content of the soil surface layer, identifing the droughts of 1858-1859, 1921-1922 and 1953-54 as extreme. However, there is no doubt, similar to flood analysis, that verifying the model results according to the actual water level and flow rate increases their credibility considerably. We have a relatively large range of palaeostage indicators to describe the maximum water levels during a flood. These palaeoflood indicators comprise various types of sedimentary (e.g. slackwater flood deposits) and botanical evidence such as impact marks and damage on trees (Benito et al., 2004, 2015, Wilhelm et al., 2019, Schulte et al. 2019). NM Lines 107-116

CA: Therefore, low water level indicators available through documentary sources are unique data records (Brazdil et al., 2018) for recording past hydrological droughts, with the precision given by physical imprints provided by epigraphic marks. During the drought, attention was paid to objects normally hidden below the water level. Most often these were large boulders, protruding rocks and sometimes even point bars or slip-off slope sandy deposits with specific local names. In many cases these were also artificial objects, protruding foundations of old bridges and building elements; around the Rhine these were the remains of old buildings or old bridges etc. (Wittman, 1859). Sometimes there was an interesting local tradition; in the sandstone area on the Czech/Saxon border it was the creation of commemorative inscriptions, particularly inscribing the current year with the low water level. Today, these objects are mostly called hunger stones.

Lines 117-119 NM:

CA: This article focuses on these hunger stones; it seeks to clarify their purpose, origin and meaning. Traditionally, water management experts and historians and perhaps ethnographers in Bohemia considered inscriptions and the year as indicated on hunger stones to be an interesting phenomenon symbolising drought.

CA: We have therefore focussed on the city of Děčín, located in the lower section of the Czech part of the Elbe river basin. The best-known hunger stone is located here and all important height surveying of all the epigraphic marks was undertaken in the summer of 2015. In 2018 the whole stone was scanned. This article discusses to what extent the inscription years have the character of historical minimum water levels.

NM

Objectives 1. To document and explain the phenomenon of hunger stones in more detail. 2. Are the year marks only commemorative for that dry year and when do they represent exact records of annual minimum water levels? 3. Are there consistent relations in the heights of stage minima among different stones? 4. What is the relation to the systematic series of measurements? 5. Do the elevations suggest any trend in water levels?

NM, lines 138-142

CA: Following the reviewer (R1) advice I changed the title The Elbe river valley between Litoměřice and Pirna was made famous by a number of prints and paintings by 19th century Romantic painters such as Adrian Zinggs (1734 – 1816) and Caspar David Friedrich (1774 – 1840). Zinngs was Swiss, but lived in Dresden; he probably coined the name of the Saxon Switzerland region, which later extended to Czech — Saxon Switzerland (Frölich – Schauseil, A., 2018). NM, lines 147-152

CA: In addition to wood, local sandstone was a traditional building and sculptural material here and throughout the North Bohemian region. However, it was also used for rich epigraphic production on the spot — on rocks and boulders (Jenč, Peša, Barus, 2008). It is quite logical that water levels were recorded adjacent to the river where possible, both minima and maxima.

CA: At the centre of our study is the city of Děčín (Fig. 1), known among other things for its unique series of flood marks (Brázdil et al., 2005, Elleder, 2016a) and hunger stone. The earlier documentation, which comes from commission inspections of the Elbe riverbed revealed previously unknown facts. In 1842, there were still a total of three hunger stones in the city of Děčín with engraved years, two on the left bank [HS1, HS3] and one on the right bank upstream of the ferry crossing [HS2] (Protokoll, 1842). The preserved stone [HS3] which is located in the lower part of the deeper riverbed is the centre of our attention. NM: lines 165-168

CA. It is related to the confluence of the Elbe River with the Ploučnice River entering from the right, the Jílovská potok stream from the left and the sediment deposits.

NM: lines 168-174

CA: This place was probably advantageous long ago as a settlement with a ford at the river confluence and below the protruding sandstone ridge. At the end of the 13th century a royal town was founded here (Fig.1, Velimská, 1991). Possibly in connection with the period of a significant occurrence of floods between 1342 and 1374 (Elleder, 2015), it was abandoned and transferred to the other side of the rock ridge, where a castle stood and the manor house is situated nowadays. There were at least two places in Děčín that were problematic from a navigational point of view. The first hunger stone [HS1] was located near the first water shallows area.

Please also note the supplement to this comment:
https://www.clim-past-discuss.net/cp-2019-113/cp-2019-113-AC1-supplement.pdf

———————————————

[Figure]

**Supplement:**

**Libor Elleder (CA): Answers to Neil Macdonald (NM)**

NM The paper is worthy of publication, however as a native English speaker it requires considerable
work to ensure that the paper is articulating the findings clearly and concisely.
I have attempted to help with the annotated comments; however, I am unable to work
through the whole paper in detail because of time constraints, this requires extensive
reworking.
1/ I would recommend reviewing the sub-heading titles and shortening them in places
e.g. 5.1 is too long.

CA The text was proofreded and by a native spakerm a professional proofreader.

NM Figure 4: it would be helpful to have a full image of the stone as
an insert depicting the four sections, and then present these four sections as you have
them - the combined image is embedded in Table 4.

CA: This is a good idea. Unfortunately, the pictures available are very similar to the upper part of the scan in Fig., 4
however due the confusing shadows and light and bad reading of the marks are not suitable for publication.

**Dear reviewer, I hope , I accepted and corrected all points you highlighted.**

[revised manuscript text omitted]

---

## Author Comment (AC2) · 16 Mar 2020

Corresponding author Libor Elleder (CA): Answers to unknown reviewer (R1)

CA: An error in the text, which I found, was corrected in the text independently from the reviewers. The beginning of Magdeburg record is 1727 not 1726 (it is an end of not monitored water level period). I am sorry for the mistake. I hope, all your comments were accepted

1/R1: Particularly on the first part of the paper, it is not very easy to read, and some sentences are too long English use may need some improvements. I cannot help on this because I am not native speaker, but I have provided some editing comments. CA: I hope, the proofreading by native speaker (It was done by a professional proof-reader

[Figure]

Erin Naillon, please see the acknowledgments) improved the English and facilitated better understanding and reading (R1: not very easy to read) of the text significantly.

2/ R1: Specific comments:

R1: Lines 97-98. Suggested modification of the sentence.

A remaining issue is to verify the credibility of the information on low water levels, and its transformation to provide robust information on runoff previous to 1825 and even before 1726.

CA: It was changed after proofreading as follows: But what credible documents of low water levels existed before 1851 (the start of record-keeping in Děčín), 1825 (the start of record-keeping in Prague) or 1727 (the start of record-keeping in Magdeburg)?

R1: Lines 104-106. Suggested change: These palaeoflood indicators comprise various type of sedimentary (e.g. slackwater flood deposits) and botanical evidences such as impact marks and damages on trees (Benito et al, 2004, 2015).

CA: I fully accepted, I change the texty accordingly (included the sugestion of Reviewer2) These palaeoflood indicators comprise various types of sedimentary (e.g. slackwater flood deposits) and botanical evidence such as impact marks and damage on trees (Benito et al., 2004, 2015, Wilhelm et al., 2019, Schulte et al. 2019).

R1: Line 106-107. However, similar methods for estimating low water levels and flow rates are seldom addressed, with some exceptions (Shamir et al., 2013). Shamir, E. et al., 2013. Geomorphology-based index for detecting minimal flood stages in arid alluvial streams. Hydrol. Earth Syst. Sci., 17, 1021-1034.

CA: I have included the citation with coments, the sentence after proofreading.

Low water levels and flow rates for preinstrumental hydrology are seldom addressed, with some exceptions. For instance, Shamir et al. (2013) presented methodology to identify field-based geomorphologic marks of low flows in ephemeral arid streams that can be indicative of minor flash floods. Unfortunately, the motivation is different and the potential for indicating historical low flows in humid climates has low utilisation.

R1: Line 107-108. Therefore, low water level indicators available through documentary sources are unique data records (Brazdil et al., 2018) for recording past hydrological droughts, with the precision given by physical imprints provided by epigraphic marks.

CA: I fully accepted, I change the texty accordingly to you

R1: Line 129 Objectives.

The specific issues and questions addressed are: Line 133. Are there consistent relations in the heights of stage minima among different stones?

CA: I fully accepted, I change the texty accordingly Are there consistent relations in the heights of stage minima among different stones?

R1: Line 136. Suggest change the title 2. The Elbe river region in the Czech Republic

CA: I fully accepted, I change the text accordingly The Elbe River region in the Czech Republic and the city of Děčín

R1: Line 145. Delete "rock"

CA: I have deleted the expression "rock", "the sandstone formation" remaine here

Below Děčín, it then flows through a landscape of sandstone formations.

R1: Line 167. Ploucnice River and Jilovsky stream should be placed on map in Figl 1
CA: Is done

R1: Line 174. Insert in brackets like this (1 ell=59 cm)

CA: I have inserted in the text: of the Prague ell units of length (1ell = 59 cm)

R1: Line 183. (see chapters on methodology and documentary sources).

CA: I modify this and replace by more suitable (see chapter 3.5.). It still bears the original German, now popular, name of 'Heger', or supervision. Later, the observation was transferred to a new water gauge [G1851] (see Chapter 3.5.).

R1: Line 196. Check if the proper word is acquired or recorded by

R1: Line 361. The brackets are confusing. Probably better as . . ...in Strahov, Wiesenfeld, C2 1844).

CA: o.k. I deleted remove the right bracket after 1643. It cannot be ruled out, for example, that mapping of the Vltava River (by David Altmann of Eidenburg) and the river regulation by Kryšpín Fuk (1640-1643), abbot of the Premonstratensian monastery in Strahov, (Wiesenfeld, 1844) were made possible just by a drier period, probably culminating in 1642 (documented by Pekař, 1998).

R1: Line 426. Rhine from 70 AD to 1858 . . . Please, check if 70AD is correct or some numbers are missing.

CA: o.k. Is really o.k. A.D.70

R1: Line 482. . . . Walter, 1901) of which reported altitudes exist for 1541, . . ..

CA: o. k.: There were a total of 10 DM marks: 1541, 1692, 1750, 1764, 1797, 1823, 1848, 1858, 1891 and 1893. Walter reported the height above sea level for the marks from 1541, 1750, 1858, 1891 and 1893.

R1: Line 494. Figure caption is confusing until the text is read. Probably change the caption as follow. Fig. 2. Drawing documenting the position of the hunger stones known as Ara Bakchi, Altarstein or Elfenstein near Bacharach, perhaps in the dry season of 1636, 1639 or 1642 (Merian, 1645).

CA: It was accepted, the caption is after proofreading, as follows

Fig. 2. Drawing documenting the position of the hunger stones known as Ara Bakchi, Altarstein and Elfenstein near Bacharach, perhaps in the dry season of 1636,1639 or 1642 (Merian, 1645), the position of which is marked by a red triangle in a cut-out view

[Figure]

of Bacharach.

R1: Line 513. . . . water levels of the Elbe River occurs typically from June to.. Line 571. Add space after between.

CA: o.k.

R1: Line 649. Perhaps you meant August 2017

CA: o.k.

R1: Line 666. In the locality opposite to Prossen. . .

CA: After proofreading: I included "to", the proof-reader excluded it, I respected the proofreading In the locality opposite Prossen village is a stone that is most often mentioned.

R1: Line 678. Marked DM minima includes years 1893..

CA: After proofreding: The marked DM includes minima: 1893, 1904, 2003 and 2018.

R1: Line 738. . . . than marking the flood mark, due to the following reasons:

CA: I accepted it fully, after proofreding: It is more difficult to make a mark of the minimum water level than to make a flood mark, due to the following reasons:

R1: Line 763. Insert in brackets the translation of the inscription, otherwise we cannot follow the meaning of the popular inscription.

CA: In connection with the 1904 mark, the popular inscription 'Wenn du mich siehst dann weine', (If you see me, you will weep) was created.

R1: Line 916. Perhaps you meant "phenological" CA: OK

Please also note the supplement to this comment:
https://www.clim-past-discuss.net/cp-2019-113/cp-2019-113-AC2-supplement.pdf

[Figure]

**Supplement:**

**Low Water Stage Marks on Hunger Stones: Verification for the Elbe River from 1616 to 2015**

Libor Elleder [1], Ladislav Kašpárek [2], Jolana Šírová, [3] and Tomáš Kabelka. [4]

[1] Applied Hydrological Research Department, Czech Hydrometeorological Institute, Prague, Czech Republic.

[2] T. G. Masaryk Water Research Institute, p. r. i., Department of Hydrology, Prague, Czech Republic.

[3] Hydrological Database and Water Balance, Czech Hydrometeorological Institute, Prague, Czech Republic.

[4] Prague Regional Office, Department of Hydrology, Czech Hydrometeorological Institute, Prague, Czech Republic.

*Correspondence to*: Libor Elleder (libor.elleder@chmi.cz)

in case of problems, send an email to  tomas.kabelka@chmi.cz

Colours rew1, rew2 denote corrections by two rewievers

[revised manuscript text omitted]

---

## Author Response (AR1)

**ANSWERS AND MARKED UP TEXT**

**Corresponding author Libor Elleder (CA):**

CA:

1) **An error in the text, which I found, was corrected in the text. The beginning of Magdeburg record is 1727 not 1726 (it is an end of not monitored water level period). I am sorry for this mistake.**

2) **The whole text has been proof red by native speaker, please see acknowledgment.**

**Answers to unknown reviewer  (R1)**

**I hope, all your commentes were accepted**

**1/R1:  Particularly on the first part of the paper, it is not very easy to read, and some sentences are too long**
**English use may need some improvements. I cannot help on this because I am not native speaker, but I have provided some editing comments.**

CA: I hope, the proofreading by native speaker improved the English and facilitated better understanding and reading (R1: not very easy to read)  of the text significantly.

2/ R1**:  Specific comments:**

**R1: Lines 97-98**. Suggested modification of the sentence.

A remaining issue is to verify the credibility of the information on low water levels, and its transformation to provide robust information on runoff previous to 1825 and even before 1726.

**CA:  It was changed after proofreading as follows**:

But what credible documents of low water levels existed before 1851 (the start of record-keeping in Děčín), 1825 (the start of record-keeping in Prague) or 1727 (the start of record-keeping in Magdeburg)?

**R1: Lines 104-106**. Suggested change: These palaeoflood indicators comprise various type of sedimentary (e.g. slackwater flood deposits) and botanical evidences such as impact marks and damages on trees (Benito et al, 2004, 2015).

**CA: I fully accepted, I change the texty accordingly (included the sugestion of Reviewer2)**
These palaeoflood indicators comprise various types of sedimentary (e.g. slackwater flood deposits) and botanical evidence such as impact marks and damage on trees (Benito et al., 2004, 2015, Wilhelm et al., 2019, Schulte et al. 2019).

**R1: Line 106-107**. However, similar methods for estimating low water levels and flow rates are seldom addressed, with some exceptions (Shamir et al., 2013). Shamir, E. et al., 2013. Geomorphology-based index for detecting minimal flood stages in arid alluvial streams. Hydrol. Earth Syst. Sci., 17, 1021-1034.

**CA: I have included the citation with coments, the sentence after proofreading.**

Low water levels and flow rates for preinstrumental hydrology are seldom addressed, with some exceptions. For instance, Shamir et al. (2013) presented methodology to identify field-based geomorphologic marks of low flows in ephemeral arid streams that can be indicative of minor flash floods. Unfortunately, the motivation is different and the potential for indicating historical low flows in humid climates has low utilisation.

**R1: Line 107-108**. Therefore, low water level indicators available through documentary sources are unique data records (Brazdil et al., 2018) for recording past hydrological droughts, with the precision given by physical imprints provided by epigraphic marks.

**CA: I fully accepted, I change the texty accordingly to you**

**R1: Line 129 Objectives**.

The specific issues and questions addressed are:
Line 133. Are there consistent relations in the heights of stage minima among different stones?

**CA: I fully accepted, I change the texty accordingly**
Are there consistent relations in the heights of stage minima among different stones?

**R1: Line 136.** Suggest change the title 2. The Elbe river region in the Czech Republic

**CA: I fully accepted, I change the texty accordingly**

**The Elbe River region in the Czech Republic and the city of Děčín**

**R1: Line 145**. Delete "rock"

**CA: I have deleted the expression „rock", „the sandstone formation" remaine here**

Below Děčín, it then flows through a landscape of sandstone formations.

**R1: Line 167**. Ploucnice River and Jilovsky stream should be placed on map in Figl 1

**CA: Is done**

[Figure]

R1: **Line 174**. Insert in brackets like this (1 ell=59 cm)

**CA: I have inserted in the text:** of the Prague ell units of length (1ell = 59 cm)

R1: **Line 183**. (see chapters on methodology and documentary sources).

**CA: I modify this and replace by more suitable (see chapter 3.5.).**

**It still bears the original German, now popular, name of 'Heger', or supervision. Later, the observation was transferred to a new water gauge [G1851] (see Chapter 3.5.).**

R1: **Line 196**. Check if the proper word is acquired or recorded by

**CA: The proofreding hase changed this sentence.**

Prof. Harlacher, the first head of the Prague Hydrological Service (Elleder, 2012), needed a long water level series for studying past drought periods. In 1875–1880 he obtained the oldest series from the Water Management Directorate in Magdeburg.

R1: **Line 361**. The brackets are confusing. Probably better as . . ...in Strahov, Wiesenfeld, C2 1844).

**CA: o.k. I deleted the right bracket after 1643.**
It cannot be ruled out, for example, that mapping of the Vltava River (by David Altmann of Eidenburg) and the river regulation by Kryšpín Fuk (1640-1643), abbot of the Premonstratensian monastery in Strahov, (Wiesenfeld, 1844) were made possible just by a drier period, probably culminating in 1642 (documented by Pekař, 1998).

R1: **Line 426**. Rhine from 70 AD to 1858 . . . Please, check if 70AD is correct or some numbers are missing.

**CA: o.k.** Is really o.k. A.D.70

Sehr trockene Zeiten waren:

**I. Vor Chrifti Geburt.**

428 in Rom.
426 große Hitze.
392 heißer Jahrgang.
390 heißer Sommer.
212 unerträgliche Hitze.
182 ungewöhnlich trockener Sommer.

**II. Nach Chrifti Geburt.**

Zwischen den Jahren 14 bis 42 werden von Dio Caffius (hist. libr. 59 cap. 7 et 23) mehrere heiße Jahrgänge erwähnt, in welchen das Forum in Rom mit Ueberhängen überspannt und das Theater in das Diribitorium (Gebäude, wo den Soldaten Sold und dem Volke Geschenke ausgetheilt wurden, bei der jetzigen Kirche S. Nicola), wo man mehr vor der Sonne geschützt war, verlegt werden mußte; auch wurde gestattet, theffalische Hüte im Theater zu tragen.

**70.** In diesem Jahre, meldet Tacitus (histor. libr. IV. cap. 26), „war der Rhein durch eine in jenem Himmelsstriche unbekannte Trockenheit kaum zur Schifffahrt tauglich, daher kärgliche Zufuhr; Wachtposten wurden längs des ganzen Ufers aufgestellt, um die

**R1: Line 482**. . . . Walter, 1901) of which reported altitudes exist for 1541, . . ..

**CA: o. k.:**
There were a total of 10 DM marks: 1541, 1692, 1750, 1764, 1797, 1823, 1848, 1858, 1891 and 1893. Walter reported the height above sea level for the marks from 1541, 1750, 1858, 1891 and 1893.

**R1: Line 494**. Figure caption is confusing until the text is read. Probably change the caption as follow. Fig. 2. Drawing documenting the position of the hunger stones known as Ara Bakchi, Altarstein or Elfenstein near Bacharach, perhaps in the dry season of 1636, 1639 or 1642 (Merian, 1645).

**CA: It was accepted, the caption is after proofreading, as follows**

Fig. 2. Drawing documenting the position of the hunger stones known as Ara Bakchi, Altarstein and Elfenstein near Bacharach, perhaps in the dry season of 1636,1639 or 1642 (Merian, 1645), the position of which is marked by a red triangle in a cut-out view of Bacharach.

R1: Line 513. . . . water levels of the Elbe River occurs typically from June to..

CA: Low water levels of the Elbe typically occur from June to September, but in 1874, for example, they lasted until December (Elleder et al., 2020)

R1: Line 571. Add space after between.

**CA: o.k.**

R1: Line 649. Perhaps you meant August 2017

**CA: o.k.**

R1: Line 666. In the locality opposite to Prossen. . .

**CA: After proofreading: I included „to ", the proof-reader excluded it, I preferred the proofreading**
In the locality opposite Prossen village is a stone that is most often mentioned.

**R1**: Line 678. Marked DM minima includes years 1893..

**tCA: This sentence after proofreding:** The marked DM includes minima: 1893, 1904, 2003 and 2018.

**R1: Line 738**. . . . than marking the flood mark, due to the following reasons:

**CA: I accepted it fully, this sentence after proofreding is as folows:**

It is more difficult to make a mark of the minimum water level than to make a flood mark, due to the following reasons:

**R1: Line 763**. Insert in brackets the translation of the inscription, otherwise we cannot follow the meaning of the popular inscription.

**CA:** In connection with the 1904 mark, the popular inscription *'Wenn du mich siehst dann weine'*, (If you see me, you will weep) was created.

**R1**: Line 916. Perhaps you meant "phenological"
**CA: Yes, is corrected.**

**Answers to reviewer 2, Neil Macdonald (NM)**

NM The paper is worthy of publication, however as a native English speaker it requires considerable work to ensure that the paper is articulating the findings clearly and concisely.
I have attempted to help with the annotated comments; however, I am unable to work through the whole paper in detail because of time constraints, this requires extensive reworking.

1/ I would recommend reviewing the sub-heading titles and shortening them in places
e.g. 5.1 is too long.

**CA  You are right, is done**

**3.6. Credibility of minimum flow marks**
**5.1 Credibility of minimum flow marks**

**CA The text has been proofred by a native speaker.**

NM Figure 4: it would be helpful to have a full image of the stone as an insert depicting the four sections, and then present these four sections as you have them - the combined image is embedded in Table 4.

CA: This is a good idea. Unfortunately, is not possible due the technical reasons, it was necessary present the drought marks large enough for reading. This is possible in the format chosen. Yes, I tried also your idea, but with small success. Yes, this solution is a compromise based on the material available.

*CA: OK*

*The greater  value, „documentary" was deleted*

[revised manuscript text omitted]